neuroscience/cognition/psychology

language lateralization, hemispheric dominance, functional transcranial Doppler sonography, structural equation modelling, handedness

**Author for correspondence:**
Z. V. J. Woodhead
e-mail: zoe.woodhead@psy.ox.ac.uk

# An updated investigation of the multidimensional structure of language lateralization in left- and right-handed adults: a test–retest functional transcranial Doppler sonography study with six language tasks

Z. V. J. Woodhead[1], P. A. Thompson[1], E. M. Karlsson[2] and D. V. M. Bishop[1]

[1]Department of Experimental Psychology, University of Oxford, Oxford, UK
[2]School of Psychology, Bangor University, Bangor, UK

ZVJW, 0000-0003-0462-6791

A previous study we reported in this journal suggested that left and right-handers may differ in their patterns of lateralization for different language tasks (Woodhead *et al.* 2019 *R. Soc. Open Sci.* **6**, 181801. (doi:10.1098/rsos.181801)). However, it had too few left-handers ($N = 7$) to reach firm conclusions. For this update paper, further participants were added to the sample to create separate groups of left- ($N = 31$) and right-handers ($N = 43$). Two hypotheses were tested: (1) that lateralization would be weaker at the group level in left-than right-handers; and (2) that left-handers would show weaker covariance in lateralization between tasks, supporting a two-factor model. All participants performed the same protocol as in our previous paper: lateralization was measured using functional transcranial Doppler sonography during six different language tasks, on two separate testing sessions. The results supported hypothesis 1, with significant differences in laterality between groups for four out of six tasks. For hypothesis 2, structural equation modelling showed that there was stronger evidence for a two-factor model in left than right-handers; furthermore, examination of the factor loadings suggested that the pattern of laterality

across tasks may also differ between handedness groups. These results expand on what is known about the differences in laterality between left- and right-handers.

# 1. Introduction

This study is an update on Woodhead *et al.* [1], 'Testing the unitary theory of language lateralization using functional transcranial Doppler sonography in adults', published in this journal. In that study, we used structural equation modelling to test the factorial structure of lateralization across six different language tasks in a group of 37 adults. Here, we report updated results from an extended sample that includes a larger group of left-handers.

It is commonly assumed that language laterality is a unidimensional trait: if an individual is strongly lateralized (relative to other individuals) on one language task, they will be strongly lateralized on all other language tasks. In structural equation modelling terms, this would be represented as one latent factor predicting covariance in laterality across all tasks. In Woodhead *et al.* [1], we tested the alternative hypothesis that laterality on one task does not predict laterality on all other tasks; instead, laterality on one set of tasks may covary independently from another set of tasks. This was represented with a structural equation model with two factors. The results supported the two-factor model, where covariance in lateralization in the tasks varied along two independent dimensions. However, further investigation of those results suggested that the two-factor model was driven by a small minority of outlier individuals, all of whom happened to be left-handed. For the majority of participants (almost all of whom were right-handed), the evidence for the two-factor model was weak. Based on these results, we speculated that there may be differences between right- and left-handed groups in their patterns of covariance in language lateralization; but as left-handers were underrepresented in the sample (the original sample contained 30 right-handers and 7 left-handers), we were unable to test this conclusively.

To test this further, we have now collected more data to increase the number of left-handed participants in the sample. Our hypotheses were:

— The left-handed group would have weaker lateralization on average than the right-handed group across all six language tasks.
— The left-handed group would show weaker covariance in lateralization between tasks than the right-handed group, with a pattern of results supporting a two-factor model of language lateralization.

The first hypothesis was based on multiple previous findings in the literature showing effects of handedness on language lateralization. It is known that lateralization does not mirror the direction of handedness, as the majority of left-handers are left-lateralized for language; but the proportion of people with bilateral or right lateralization is higher in left-handers than right-handers [2–6]. We aimed to replicate this finding and to extend it by exploring differences between handedness groups across a range of different language tasks.

A meta-analysis by Carey & Johnstone [6] studied the effects of handedness on various measures of language laterality, including the likelihood of aphasia from left or right hemisphere stroke; behavioural measurements such as dichotic listening or visual half-field tasks; and (of particular relevance for this study) functional imaging methods such as functional transcranial Doppler sonography (fTCD) or functional magnetic resonance imaging (fMRI). The results showed significantly greater left hemisphere dominance in right- compared with left-handed participants, across all paradigms. In functional imaging experiments with healthy controls, this has been reported for a wide variety of language: most commonly for speech production tasks such as word generation (i.e. verbal or phonological fluency) [4,7–14]; verb generation [15]; and animation description [16]; but also for a semantic decision with auditory stimuli [17]. Conversely, no effect of handedness was seen in a study of passive listening with overlearned speech sequences (the months of the year) [18]. Hence, while there is ample evidence that handedness *does* affect language lateralization, it is as yet unclear whether this can be seen for all language functions, or just some. In this study, we aimed to test the effects of handedness on laterality using six different language tasks, covering a range of functions including speech generation (propositional and automatic), speech comprehension, semantics, phonology and syntactics.

Our second hypothesis was prompted by the pattern observed in the small number of left-handed participants ($N = 7$) in the previous paper. We previously observed that when a mixed group of left- and right-handers was entered into the model, the data supported a two-factor model of covariance

**Table 1.** Number of participants included in the final analysis by handedness group (left or right) and site (Oxford or Bangor).

|  | left-handers | right-handers | all |
| --- | --- | --- | --- |
| Oxford | 7 | 30 | 37 |
| Bangor | 24 | 13 | 37 |
| all | 31 | 43 | 74 |

between tasks, but this was driven in part by the inclusion of individuals who were multivariate outliers, all of whom were left-handed. In the present study, we aimed to explore whether there is stronger evidence for two independent factors of language lateralization in left-handers compared with right-handers; and if so, whether the two handedness groups showed different patterns of covariance across tasks. No other studies, to our knowledge, have looked at whether left- and right-handed participants differ in their pattern of lateralization across language tasks.

# 2. Material and methods

## 2.1. Preregistration and data sharing

The initial project reported in Woodhead et al. [1] was preregistered on Open Science Framework (OSF) (https://osf.io/9uaw4/), but the additional data collection and updated analysis was not. Many aspects of the data analysis used in this updated project remained the same as in the initial preregistration. Any changes to the analysis procedure are noted below. The new data and analysis scripts for the updated project are also available on OSF (https://osf.io/qetj5/).

## 2.2. Design

The design was the same as that of the initial project. Lateralization of brain activity during six language tasks was measured in a test–retest design using functional transcranial Doppler sonography (fTCD). Unlike the initial project, participants were grouped according to handedness, creating a mixed design with a between-subjects variable of handedness (left or right) and a within-subject variable of session (1 or 2).

The two sessions were spaced by between 3 days and 6 weeks. Hence, the raw data for each participant comprised fTCD recordings for the six tasks (labelled A–F) tested in two sessions (A1–F1, A2–F2).

The initial data collection was performed at the University of Oxford. The additional data included in the sample reported here were collected at Bangor University.

## 2.3. Participants

All participants gave written informed consent. Procedures were approved by the local site (University of Oxford Medical Sciences Interdivisional Research Ethics Committee, or Bangor University School of Psychology Ethics Committee). Participants (mostly students) were recruited via local subject panels or poster advertisements. The inclusion criteria were the same at both sites: participants were aged 18–45 years, spoke English as their first language and had normal or corrected to normal hearing and vision. The exclusion criteria were a history of significant neurological disease, head injury or developmental language disorder.

The sample size in the initial project was 37 (30 right-handed and 7 left-handed participants). To allow us to investigate the effect of handedness, we aimed to increase the sample to at least 30 participants in each group. In order to mitigate the potential confound of testing site, we also aimed to test at least 10 right-handed participants at the new site (Bangor).

A total of 43 participants were recruited in Bangor. Data collection for five of these participants was not completed, as the participant either failed to return for the second testing session, or because it was not possible to identify a clear Doppler signal from the middle cerebral arteries. Hence, in Bangor, full datasets were collected from 38 participants (13 right-handed, 25 left-handed). During the data analysis, one left-handed participant from Bangor was excluded due to poor data quality.

The extended data reported here, combining participants recruited at Oxford or Bangor, comprised 74 participants (43 right-handed, 31 left-handed). Of these, 47 were female and 27 were male. Handedness was

assessed using a self-report question with three options: left-handed, right-handed or ambidextrous. No participant selected 'ambidextrous'. The breakdown of participant handedness by site is reported in table 1.

The participants' mean age was 23.7 years (s.d. = 5.6 years); for left-handers, the mean was 24.1 years (s.d. = 5.7) and for right-handers, it was 23.2 years (s.d. = 5.5).

To address a potential concern of language differences between the two sites, we administered a language proficiency questionnaire. All participants reported English to be their first language. We looked at how many participants started learning a second language from a young age (3 years or younger). In Oxford, there were three such participants (one in Persian, one in Chichewa and one in French), and in Bangor, there were two (one in Welsh and one in Polish). We also looked at the mean number of languages spoken (to any level of proficiency). This was similar across sites: 2.45 languages in Oxford and 2.67 language in Bangor. Hence, the two samples seemed well matched for their foreign language experience. Note that a previous study showed no difference in language laterality (using fTCD) for first and second languages in proficient bilinguals [19].

## 2.4. Procedure and language tasks

All six tasks were administered at each session, and the order of the tasks was counterbalanced between participants.

The procedures and rationale for the six tasks are described in detail in our previous paper [1]. The tasks were chosen to tap a broad range of language functions, encompassing production, perception, phonology, semantics and syntax. In brief, the tasks were:

A. List Generation, which required production of automatic speech (counting, reciting the days of the week or months of the year) in response to a picture
B. Phonological Decision, where participants decided whether the names of two pictures rhymed
C. Semantic Decision, where participants decided whether two pictures were semantically related
D. Sentence Generation, which required production of a meaningful sentence to describe a picture
E. Sentence Comprehension, where participants decided which of two pictures matched a spoken sentence
F. Syntactic Decision, where participants decided whether a sequence of words and non-words formed a plausible 'jabberwocky' sentence with correct syntactic structure.

All stimulus materials for the tasks are available on OSF (https://osf.io/8s7vn/). There were 15 trials of each task per session, administered in separate runs. All tasks shared a common structure with an inter-stimulus interval of 33 s. Trials started with a 3 s 'Clear Mind' prompt, followed by the language task for 20 s, and ended with 10 s of rest.

## 2.5. Behavioural analysis

Behavioural data from the six language tasks were analysed to check for differences in task performance between the left- and right-handed groups. For tasks A and D, the outcome measure was the average number of words spoken per trial; for tasks B, C, E and F, the outcome measures were percentage accuracy and average reaction times (RTs) for correct trial (excluding incorrect answers and RTs more than 2 s.d. away from the mean). The number of trials where participants failed to respond was also recorded (accuracy on these trials was scored as incorrect).

## 2.6. Functional transcranial Doppler sonography analysis

Simultaneous bilateral fTCD was used to measure the cerebral blood flow velocity (CBFV) in the left and right middle cerebral arteries (a proxy measure for brain activity). The middle cerebral artery territory covers the lateral temporal, inferior frontal and inferior parietal lobes [20], so it is assumed that blood flow here will be sensitive to activity in most language-related brain regions [21]. In the fTCD analysis, the CBFV during the language task is compared with the resting baseline between trials, then the difference between left and right CBFV is taken to produce a laterality index (LI). Hence, the dependent measure produced by the analysis is an LI for each participant, for each task, at each session.

The fTCD data were analysed using a custom script in R Studio [22], which is available on OSF (https://osf.io/qetj5/). The process for calculating LI from the raw fTCD data was identical to that described in the initial paper [1]. The analysis steps included:

— down-sampling to 25 Hz;

— epoching into individual trials from −7 to 27 s peri-stimulus time;

— removing trials with multiple CBFV values outside of an acceptable range;

— manually removing trials with other obvious artefacts or no behavioural response;

— data normalization;

— heart cycle integration;

— baseline correction using a baseline period from −5 to 2 s peri-stimulus time;

— removing trials containing values below 60% or above 140% of the mean normalized data range;

— averaging of data across trials;

— calculating the difference between averaged left and right CBFV;

— calculation of the LI by taking the mean difference in CBFV within the period of interest when the participant was performing the language task. This method was used instead of the more commonplace method of identifying the peak in the CBFV difference, as we have previously demonstrated that the peak method can induce artificial bimodality in the LI data [23].

The number of useable trials for each participant and task was summed. If a participant had less than 10 useable trials for a particular task, their data for the whole task were excluded from the analysis. This was a modification from the previous analysis, where the criterion was 12 useable trials, because it seemed reasonable to include those with a small number of trials provided trial-by-trial variability for a task was low. As described in the previous paper, we checked this by rejecting data points where the standard error of the LI value from trial to trial was unusually high using Hoaglin & Iglewicz's procedure [24]. If more than one task was excluded for a participant, that participant's dataset was excluded entirely. One left-handed participant from Bangor was excluded for this reason.

## 2.7. Structural equation modelling

We used structural equation modelling (SEM) to perform confirmatory factor analysis in the OpenMx (https://openmx.ssri.psu.edu/) [25] and umx [26] packages in R. A brief description of SEM can be found in the previous paper [1]. As our previous paper had obtained good fit for a two-factor model, here we used confirmatory factor analysis, using the same two-factor model, to ask whether left- and right-handers have different patterns of covariance in language laterality across the six tasks. We tested this in two steps:

(1) Testing factorial structure: we analysed the left- and right-handed groups separately, to test whether covariances between laterality on the six language tasks were best modelled by a single- or two-factor model in each group.
(2) Testing factor loadings: as step 1 demonstrated that a two-factor model was optimal for both left- and right-handed groups, we used multigroup modelling to set up a model with two factors, incorporating both left- and right-handed groups. The aim of this was to test whether the pattern of factor loadings within the two-factor model was significantly different between left- and and right-handers.

### 2.7.1. Step 1: testing factorial structure

We analysed the left- and right-handed groups separately in this step, to see whether a single-factor model or a two-factor model was an optimal fit. All models shared a common structure, which was identical to the models of covariance reported in the previous paper [1] (figure 1). The analysis works by comparing the covariance matrix observed from the data with the covariance matrix implied by the structural model. Good fit is obtained when the observed pattern agrees well with model predictions.

Laterality indices from the six tasks at the two sessions (A1–F1, A2–F2) were used as indicators in the model. Each indicator has two free (estimated) parameters: a mean and a residual variance around that mean (indicated by the circular arrow). Following our previous paper, we fixed the means and variances to be the same at each session—i.e. the parameter for the mean of task A at session 1 was set to be the same at session 2. This feature of the model was arrived at by demonstrating that a model where means (and variances) were different for the two sessions was not significantly better than the simpler model where means (and variances) were fixed at the two sessions. This reflects the fact that the laterality indices we measured generally had good test–retest reliability.

The ovals in figure 1 are factors, or latent variables. In SEM, latent variables are not observed directly, but estimated from the observed variables (also known as the indicators or manifest variables). A well-known example is that '*g*' is a latent variable that is a representation of intelligence as measured in multiple

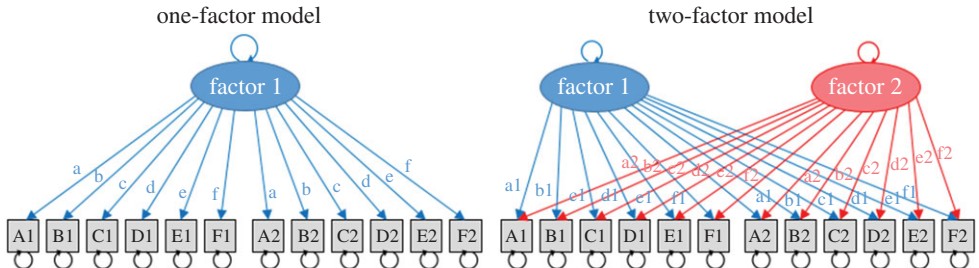

**Figure 1.** Diagrams for the SEM models tested in this analysis. The letters A–F denote the task, and the numbers 1–2 denote the session. By convention, factors are shown in ovals, and factor loadings are shown with arrows labelled with lower case letters. Where the same letter is used for two paths, they are set (or 'fixed') to be equal. The circular arrows represent residual variance around the factors or task means. The task means were included in the models, but have been omitted from the model diagrams for simplicity as our analyses focused on comparing the covariance structure (i.e. the one-factor model versus the two-factor model).

different tasks (the indicators). The factor structures determine the covariance between the indicators. The models tested had either one or two factors. To ensure the identifiability of the models when estimating parameters, we follow the usual convention of fixing certain parameters [27]. The parameters for the factors' means were fixed at zero. The parameter for factor 1's variance was free to vary, and the parameter for factor 2's variance was fixed at one. Missing data were dealt with using the full-information maximum likelihood (FIML) estimation and the raw data were input into the model rather than correlation or covariance matrices. (Note: correlation matrices for the two groups are presented in figure 3.)

The paths linking the factors and the indicators (shown in straight arrows and lower case letters) represent the estimated factor loadings. Like the task means and variances, the factor loading parameters were constrained to be the same for data from session 1 and session 2. In order to achieve model identification, it is necessary to constrain the factor loading for one of the indicators (known as the marker or reference indicator) to one. It is good practice to use the most reliable scores as the reference indicator [27]. We fixed the factor loadings with task D (Sentence Generation) as it previously showed the strongest lateralization and test–retest reliability. Hence, the factor loading from factor 1 to task D was set to one, and the loading from factor 2 to task D was set to zero. All other factor loadings were free parameters.

The one-factor and two-factor models were fit in the left- and right-handed groups separately. Model comparison was performed (for each group separately) using a likelihood ratio test to determine whether the more complex two-factor model was a significantly better fit to the data than the one-factor model.

In order to test the robustness of the winning model's parameter estimates (i.e. whether the estimates were dependent on any particularly influential participants), we repeated the model fitting and comparison (as described above) in a leave-one-out validation analysis.

### 2.7.2. Step 2: testing factor loadings

The previous step identified that the two-factor model was a better fit to the data than the one-factor model in both left- and right-handed groups. In the next step, a two-factor multigroup model was fitted to the full dataset containing both groups to test whether the left- and right-handed groups differed in terms of how the six tasks loaded onto the two laterality factors. In SEM, factor loadings are the linear regression coefficients, which are estimated for the connections between each indicator (in this case, each task) and each factor. As described above, the factor loading for one indicator has to be fixed in order to give the factors an interpretable scale: in our model, task D (Sentence Generation) was fixed to have a factor loading of 1 for factor 1, and 0 for factor 2. Factor loadings for the other tasks were estimated from the data.

Here, we tested whether the factor loadings estimated for the left- and right-handed groups were the same, or different. The purpose was to determine whether the same causal relationships between the factors holds in different groups, and if not, where the systematic differences lie. In other words, is there evidence to suggest that left- and right-handers have a different pattern of covariance across language lateralization for our six tasks?

To test this, we used the same model structure as shown in figure 1 (the two-factor model). Multigroup modelling was used so that we could estimate parameter estimates for each group simultaneously, following the same factor structure and within one fitting procedure. The advantage

of this approach (rather than modelling the groups separately) was that differences between groups could be tested explicitly.

In order to test whether the pattern of factor loadings differed between the two groups, we set up two multigroup models: a 'constrained loadings' model, where the factor loadings were fixed to be the same in both groups; and an 'unconstrained loadings' model, where they were allowed to differ between groups. As before, the two models were compared using a likelihood ratio test between the simpler model (with constrained loadings) and the more complex model (with unconstrained loadings) to see whether increasing model complexity significantly improved the model fit.

Finally, in order to visualize the extent of individual variability in the left- and right-handed groups, we computed 'weighted LI means', as a weighted mean score for each factor from all tasks, using factor loadings as weights. This corresponds to what DiStefano et al. [28] termed a 'non-refined' method of extracting factor scores. These weighted LI means, which are on the same scale as the original raw laterality indices, were plotted in a scatterplot.

# 3. Results

All data are available on OSF (https://osf.io/qetj5/).

## 3.1. Behavioural results

We collected behavioural data on the language tasks during fTCD recording. The behavioural measures included: the number of words spoken per trial (in tasks A and D); the percentage of accurate responses (in tasks B, C, E and F); the mean reaction time (for tasks B, C, E and F) and the number of omitted responses (in all tasks). These results are reported for the left- and right-handed groups in table 2. It should be noted that the reaction times for task F were particularly fast because the participants had to withhold their response until the end of the word sequence, by which time they were ready to respond immediately.

We wished to rule out the possibility that any differences in lateralization between handedness groups were due to differences in task performance. To check for this possible confound, we performed two-sample $t$-tests on the measures of the number of words spoken, accuracy and reaction time. There were 10 measures, each tested at two sessions, so a total of 20 comparisons were performed. $\alpha$ was Bonferroni-corrected and set to 0.0025 (0.05/20). No measure showed a significant effect of handedness at this level.

To test whether there were differences in task performance between sessions, we performed repeated-measures Wilcoxon rank tests on those 10 measures. All participants were included in this analysis. Using Bonferroni correction for the 10 comparisons ($\alpha = 0.005$), five measures showed significant test–retest improvements: accuracy and RT in task B (Phonological Decision; $p < 0.001$ in both); RT in task C (Semantic Decision; $p < 0.001$) and accuracy and RT in task E (Sentence Comprehension; accuracy, $p = 0.001$; RT, $p < 0.001$).

## 3.2. Lateralization results

In the fTCD analysis, trials were excluded if the participant failed to respond, gave an inappropriate response (i.e. performing the wrong task) or if there were obvious artefacts in the data. Only 4.12% of trials were excluded for these reasons.

Five individual LI values (i.e. for a specific participant, session and task) were excluded as there were less than 10 useable trials; a further five were excluded as the trial-by-trial standard error was too high (using the Hoaglin & Iglewicz [24] method). As described in §2.6, all of the data from one subject (a left-handed participant from Bangor) was removed from the analysis as they had two LI values excluded for these reasons.

The Shapiro–Wilk tests were performed to test the normality of the LI value distributions for each task, session and group (24 tests in total). Only one test showed a significant result (indicating non-normality): this was for task B1 (Phonological Decision, session 1) in left-handers ($p = 0.016$). Overall, however, the data supported the assumption of normality, and parametric tests were used throughout the remaining analysis. Density and Q–Q plots can be seen in the electronic supplementary material.

Figure 2 shows a pirate plot of the LI values for all tasks, at both sessions, in both handedness groups (omitting excluded data). This plot was created using the R package ggpirate [29]. Filled symbols denote

**Table 2.** Summary statistics of the behavioural measures from the six language tasks. For each measure, a group mean is reported, and the standard deviation is shown in parentheses.

| task | measure | session | group mean (s.d.) | |
|---|---|---|---|---|
| | | | left-handers | right-handers |
| A | words spoken | 1 | 9.31 (0.77) | 9.63 (1.08) |
| | | 2 | 9.36 (0.98) | 9.64 (0.70) |
| | omitted responses (%) | 1 | 0.00 (0.00) | 0.62 (1.96) |
| | | 2 | 0.00 (0.00) | 0.16 (1.02) |
| B | accuracy (%) | 1 | 89.67 (6.66) | 90.35 (5.56) |
| | | 2 | 91.86 (5.29) | 92.54 (4.38) |
| | reaction time (s) | 1 | 1.61 (0.19) | 1.65 (0.21) |
| | | 2 | 1.45 (0.19) | 1.49 (0.20) |
| | omitted responses (%) | 1 | 1.50 (1.91) | 2.20 (2.56) |
| | | 2 | 0.47 (1.28) | 0.77 (1.22) |
| C | accuracy (%) | 1 | 93.48 (4.68) | 95.23 (3.26) |
| | | 2 | 94.41 (4.37) | 94.53 (3.67) |
| | reaction time (s) | 1 | 1.04 (0.15) | 1.14 (0.21) |
| | | 2 | 0.97 (0.14) | 1.06 (0.19) |
| | omitted responses (%) | 1 | 0.36 (0.67) | 0.72 (1.86) |
| | | 2 | 0.18 (0.50) | 0.62 (1.20) |
| D | words spoken | 1 | 9.26 (1.38) | 9.07 (0.95) |
| | | 2 | 9.42 (1.58) | 9.14 (1.08) |
| | omitted responses (%) | 1 | 0.43 (1.67) | 0.31 (1.42) |
| | | 2 | 0.00 (0.00) | 0.31 (1.42) |
| E | accuracy (%) | 1 | 90.89 (5.57) | 91.04 (5.23) |
| | | 2 | 91.65 (4.89) | 93.51 (4.31) |
| | reaction time (s) | 1 | 2.13 (0.13) | 2.16 (0.14) |
| | | 2 | 2.10 (0.16) | 2.10 (0.16) |
| | omitted responses (%) | 1 | 2.01 (1.87) | 3.00 (2.88) |
| | | 2 | 1.18 (1.49) | 1.70 (1.54) |
| F | accuracy (%) | 1 | 84.52 (9.08) | 86.92 (9.80) |
| | | 2 | 85.52 (9.80) | 86.43 (10.95) |
| | reaction time (s) | 1 | 0.32 (0.10) | 0.34 (0.08) |
| | | 2 | 0.31 (0.08) | 0.33 (0.06) |
| | omitted responses (%) | 1 | 1.65 (2.63) | 4.13 (5.72) |
| | | 2 | 1.00 (1.50) | 4.34 (5.96) |

individual cases where the 95% confidence interval around the estimate did not include zero, i.e. these individuals were reliably lateralized to left (LI value > 0) or right (LI value < 0). Unfilled symbols correspond to individuals who were not reliably lateralized. A table of the mean LI values can be found in the electronic supplementary material. A further pirate plot is provided in the electronic supplementary material, showing data acquired at Oxford or Bangor.

One-sample $t$-tests were used to test for significant lateralization at the group level (LI $\neq$ 0). Whereas right-handers showed significant left lateralization for tasks A, B, C, D and E at both sessions, for left-handers, this was only shown for tasks A and D (the two speech production tasks). Task F (Syntactic Decision) did not show significant lateralization in either group. Task D (Sentence Generation) showed

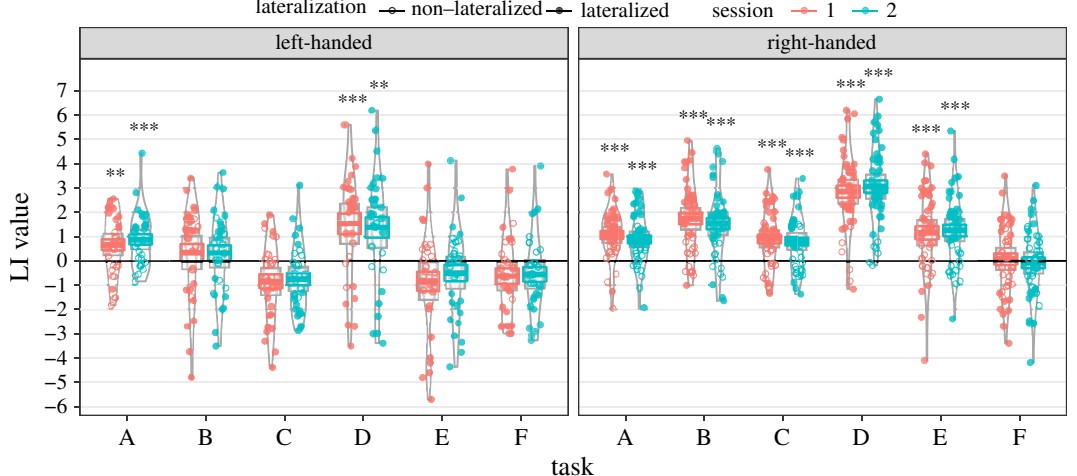

**Figure 2.** Pirate plot showing the laterality indices (LI values) for left- ($N = 31$) and right-handed ($N = 43$) groups, across all tasks (A–F) and both sessions. The coloured dots represent LI values for each participant, with filled symbols denoting individuals where the 95% confidence interval around the estimate did not include zero. The coloured lines indicate the group means; the grey boxes indicate the 95% confidence interval for the group estimate of LI; and the grey violins indicate the densities. The black asterisks show whether the group LI values were significantly different to zero using one-sample $t$-tests ($^* p < 0.05$; $^{**} p < 0.01$; $^{***} p < 0.001$). A, List Generation; B, Phonological Decision; C, Semantic Decision; D, Sentence Generation; E, Sentence Comprehension; F, Syntactic Decision.

the strongest lateralization in both groups. The plot also shows that the LI values at sessions 1 and 2 were similar for all tasks.

The pirate plot also shows that LI values for left-handers were generally lower than for right-handers across all tasks. To test this formally, we averaged LI values over sessions 1 and 2, then performed between-subject $t$-tests to compare LI between left- and right-handed groups for each task. As six comparisons were performed, $\alpha$ was Bonferroni-corrected and set to 0.0083 (0.05/6). Significant differences between left- and right-handers were seen for task B (Phonological Decision: $t_{53.88} = 3.43$, $p = 0.0012$); task C (Semantic Decision: $t_{53.07} = 5.32$, $p < 0.001$); task D (Sentence Generation: $t_{49.96} = 3.17$, $p = 0.0026$) and task E (Sentence Comprehension: $t_{52.48} = 4.52$, $p < 0.001$). The difference between groups was not significant for task A (List Generation: $t_{61.10} = 0.88$, $p = 0.385$) or task F (Syntactic Decision: $t_{58.28} = 1.60$, $p = 0.115$).

Figure 3 shows the correlation matrices of the LI values for the left- and right-handed groups. Pearson's correlations were used. In general, correlations were stronger in the left-handed group than the right-handed group. Scatterplots of the test–retest data (shown in figure 4) suggested that this may have been because of the greater inter-subject variability in the left-handed group than the right-handed group. Task A (List Generation) showed low test–retest correlations and low correlations with other tasks. The test–retest correlations for tasks B–F were medium to strong, ranging from 0.52 to 0.86. Task D had the strongest test–retest correlations in both groups. Split-half reliability data (i.e. the correlation between LI values calculated using odd versus even trials) are available in the electronic supplementary material.

## 3.3. Structural equation modelling results

### 3.3.1. Step 1: testing factorial structure

The one- and two-factor models shown in figure 1 were tested in the left- and right-handed groups separately. The model fit statistics are shown in table 3.

In the left-handed group, the fit indices were all substantially better for the two-factor model. The comparative fit index for the one-factor model was particularly poor (0.727): a cut-off criterion of 0.90 is generally agreed to indicate 'acceptable' goodness of fit [30]. As there was a large difference in the log-likelihoods between the two models (75.33), the likelihood ratio test was highly significant ($p < 0.001$). The AIC weight-based conditional probabilities [31] of the one-factor or two-factor models being the best fit to the data were '0' and '1', respectively. A leave-one-out validation analysis was run, dropping one of the left-handed participants in each iteration. In every iteration, the likelihood

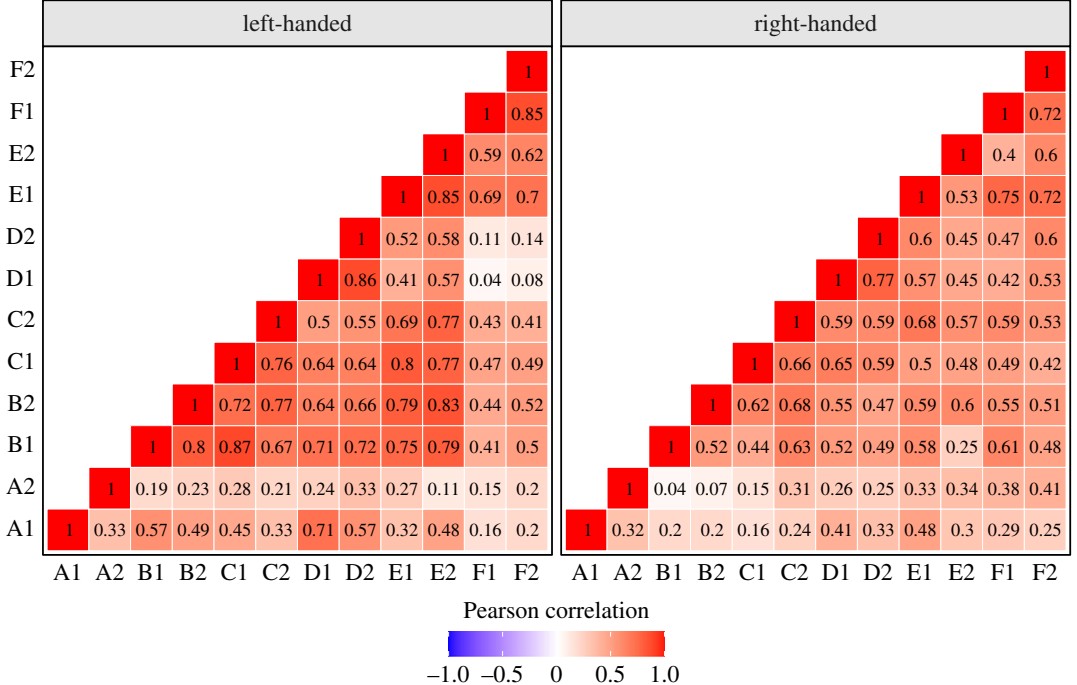

**Figure 3.** Correlation matrices for the LI values in all tasks (A–F) and sessions (1–2) for the left-handed and right-handed groups. A, List Generation; B, Phonological Decision; C, Semantic Decision; D, Sentence Generation; E, Sentence Comprehension; F, Syntactic Decision.

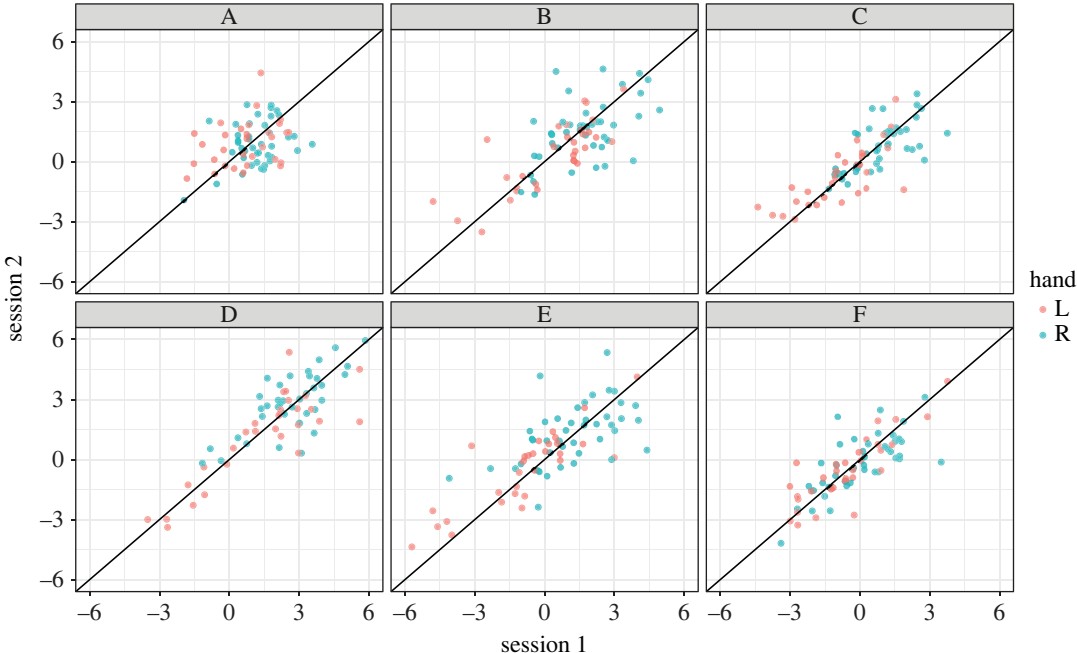

**Figure 4.** Scatterplots showing the relationship between laterality indices at session 1 (x-axis) and session 2 (y-axis) for each task. The diagonal line is the theoretical line of equivalence, where laterality in session 1 equals laterality in session 2. The distribution of points along the diagonal line indicates how much individuals varied in the strength of laterality; the spread of points perpendicular to the diagonal line indicates how variable individuals were from session to session. The colour coding indicates the participants' handedness (red, left-handed; blue, right-handed). A, List Generation; B, Phonological Decision; C, Semantic Decision; D, Sentence Generation; E, Sentence Comprehension; F, Syntactic Decision.

ratio test showed a significantly better fit for the two-factor model than the one-factor model ($p < 0.001$ in every iteration).

In the right-handed group, the fit indices still favoured the two-factor model, but the difference in log-likelihoods was smaller (19.01, $p = 0.002$); this is particularly evident in the difference in AIC (9.01), a metric

**Table 3.** Model fit statistics for the one- and two-factor models estimated for the left-handed and right-handed groups separately; and the constrained and unconstrained two-factor multigroup models. $N$ params, number of parameters; −2logL, −2 log-likelihood; d.f., degrees of freedom; AIC, Akaike's information criterion; CFI, comparative fit index; RMSEA, root mean square error of approximation.

| group | model | $N$ params | −2 logL | d.f. | $p$-value | AIC | BIC | CFI | RMSEA |
|---|---|---|---|---|---|---|---|---|---|
| left | 1 factor | 18 | 1204.239 | 351 | — | 502.24 | −1.09 | 0.727 | 0.204 |
|  | 2 factor | 23 | 1128.909 | 346 | $7.93 \times 10^{-15}$ | 436.91 | −59.25 | 0.933 | 0.105 |
| right | 1 factor | 18 | 1555.086 | 493 | — | 569.09 | −299.19 | 0.886 | 0.102 |
|  | 2 factor | 23 | 1536.077 | 488 | 0.0019 | 560.08 | −299.39 | 0.936 | 0.079 |
| multigroup | constrained | 36 | 2688.054 | 844 | — | 1000.05 | −940.47 | 0.913 | 0.071 |
|  | unconstrained | 46 | 2664.986 | 834 | 0.0105 | 996.99 | −924.60 | 0.934 | 0.064 |

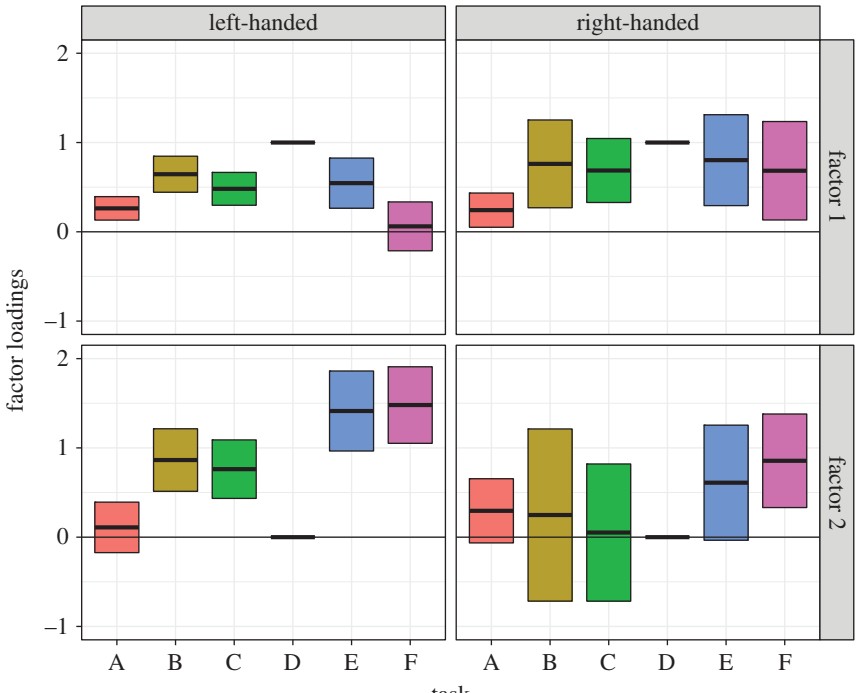

**Figure 5.** Factor loadings (and 95% confidence intervals) for left- and right-handers from the unconstrained multigroup model. Note that in order to give the factor loadings an interpretable scale, the loading for task D was fixed to 1 for factor 1, and to 0 for factor 2, and hence do not have confidence intervals. For the other tasks, confidence intervals that span zero indicate that the loading is likely to be non-significant. Tasks: A, List Generation; B, Phonological Decision; C, Semantic Decision; D, Sentence Generation; E, Sentence Comprehension; F, Syntactic Decision.

that is corrected for model complexity. The AIC weight-based conditional probabilities [31] of the one-factor or two-factor models being the best fit to the data are 0.01 and 0.99, respectively. In this group, the leave-one-out analysis only showed a significant benefit of the two-factor model in 27 out of 43 iterations.

Hence, while both handedness groups showed stronger evidence in favour of the two-factor model, this evidence appeared weaker in the right-handed group.

### 3.3.2. Step 2: testing factor loadings

In the second step, we combined data from both groups into one multigroup model. As both groups (separately) showed better model evidence for the two-factor model, we used this model in the multigroup model. Two models were evaluated: the first was constrained so that the factor loadings were fixed to be the same in both groups; and the second was unconstrained, allowing the factor loadings to differ between groups. The model fit statistics are shown in table 3. The likelihood ratio test showed that the model with unconstrained factor loadings was significantly better than the constrained model (difference in log-likelihood = 23.07, $p = 0.0105$). The other fit indices generally supported this, apart from the BIC which was lower (indicating a better fit) for the constrained model. The AIC weight-based conditional probabilities [31] of the constrained or unconstrained models being the best fit to the data were 0.18 and 0.82, respectively.

The factor loadings from the unconstrained model are shown in figure 5. This depicts how the pattern of factor loadings differed between the left- and right-handed groups. For factor 1, the loadings are quite similar for the left- and right-handed groups, except the task F (Syntactic Decision) had a very weak loading in the left-handers and a stronger loading in the right-handers. For factor 2, the left-handed group had positive loadings for tasks B, C, E and F, whereas the right-handers only had positive loadings for task F.

In order to visualize the variability between participants, weighted LI means from the unconstrained model were calculated for each individual and plotted in a scatterplot (figure 6). As described in the Material and methods, the weighted LI means are calculated for each individual by taking a weighted average of the LI values, using the factor loadings as weights. As such, positive scores indicate overall left lateralization for that factor, and negative scores indicate right lateralization.

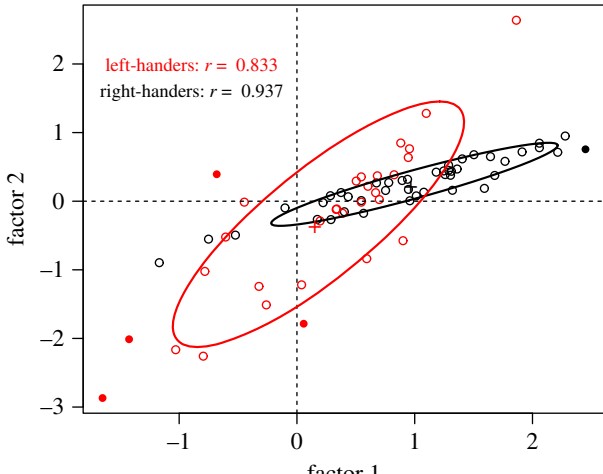

**Figure 6.** Weighted LI means for left-handers (red) and right-handers (black) from the unconstrained multigroup model. Covariance ellipses are shown for each group. Data points shown with filled circles represent outliers that were over four times Cook's distance from the group average.

As the weighted LI means were free to vary between groups in this model, group differences can also be observed in the weighted LI means. Participants in the right-handed group showed some individual variability in the weighted LI means for factor 1, but weighted LI means for factor 2 had little individual variability. This reflects the weak evidence for the two-factor model in right-handers, as observed in step 1 of the analysis. Weighted LI means from the right-handed participants clustered closely (indicated by the narrow covariance ellipse and high correlation, $R = 0.94$). Only one outlier was observed (with weighted LI means more than four times Cook's distance away from the group mean), who had an unusually high score for factor 1.

By contrast, the left-handed group had lower weighted LI means than the right-handed group on average (indicating weaker lateralization) for both factor 1 and factor 2. They also showed greater individual variability in weighted LI means for both factors, indicated by the broader covariance ellipse and the lower correlation ($R = 0.83$). Four outliers were observed: two who had unusually low weighted LI means for factors 1 and 2; and two who had unusually inconsistent weighted LI means for the two factors (i.e. fell in the top left and bottom right quadrants of the data).

# 4. Discussion

These results supported our two hypotheses: (1) lateralization was weaker in the left-handed group than the right-handed group; and (2) the left-handed group showed weaker covariance in lateralization between tasks than the right-handed group.

Focusing first on hypothesis 2, we observed that the evidence in support of a two-factor model of covariance was strong in the left-handed group, and stable in a leave-one-out validation analysis. This supports our hypothesis that, for left-handers, not all language tasks show correlated laterality indices. Instead, there was evidence that the tasks covaried in two factors with distinct patterns, as seen in figure 5.

By contrast, in right-handers, the model comparison still supported the two-factor model, but the evidence was smaller and it was not stable in the leave-one-out analysis. This instability was due to the fact that the model evidence supporting the two-factor model was weak. Figure 5 shows the fit of the two-factor model (from the unconstrained multigroup analysis). For the right-handers, the confidence intervals around the factor loadings were large, reflecting the poor fit of the model, and the patterns for factor 1 and factor 2 were less distinct than for the left-handers. Note that using the SEM approach, it is not possible to directly compare the strength of evidence for the two-factor model between left- and right-handers: model comparison is only valid for nested models, where the data remain the same but the model changes.

A direct comparison of the factor loadings in left- and right-handers was performed in the second step of the analysis, using multigroup modelling. A multigroup model with two factors was

estimated, within which it was possible to test whether model parameters differed between the two handedness groups. This analysis showed that the pattern of factor loadings differed significantly between the left- and right-handers.

The biggest differences in the factor loadings between groups (as shown in figure 5) were observed in the Semantic Comprehension and Syntactic Decision tasks (E and F). Both of these tasks required receptive sentence level comprehension. Hence, a speculative hypothesis for future work is that in left-handers (or a subset of individuals, the majority of whom are left-handed), the language processes involved in phonology, semantics and expressive speech co-lateralize together, but sentence comprehension can lateralize independently. The critical language process in tasks E and F that drove this second factor may have been speech perception (as both used auditory stimuli) or it may have been related to sentence level semantics or syntax. One quite consistent finding from the literature is that speech production is more strongly lateralized than comprehension [32–37], cf. [38], but this does not necessarily imply that those two functions lateralize independently—despite having different means, they may still be correlated. In our previous paper, we motivated task selection using Hickok & Poeppel's dual stream model of speech perception [39], which proposes a division of labour between dorsal areas involved in sensorimotor processing of speech, and ventral areas involved in lexical access and comprehension. However, although it might be tempting to identify factor 1 with the dorsal stream, and factor 2 with the ventral stream, that would not account for the loading of the semantic decision task on factor 1. A more parsimonious explanation might be that processes relating to sentence level comprehension (required for both tasks E and F), such as combinatorial semantics, syntax or working memory, may have driven the dissociation of factor 2 in at least some participants. Another possibility, which we cannot rule out, is that the difference between factors was not linguistic in nature, but driven by specific cognitive requirements of the tasks, e.g. the extent to which they involve executive functions, or simply the overall cognitive demand of the task.

With regard to hypothesis 1, we replicated previous findings by showing that language lateralization was generally weaker in the left-handers than the right-handers (at the group level). This was observed for four of the six tasks: Phonological Decision, Semantic Decision, Sentence Generation and Sentence Comprehension. Interestingly, no differences between groups were observed for List Generation (which also had particularly weak test–retest reliability in both groups) or for Syntactic Decision (which did not lateralize significantly in either group). When interpreting the differences we observed between left- and right-handers, it is important to note that in previous studies, 'atypical' lateralization has only been observed in a minority of left-handed people. For example, Mazoyer et al. [3] used Gaussian mixture modelling to analyse laterality indices from a sentence generation task acquired with fMRI. They observed that the majority of individuals (including right- and left-handers) showed strong left lateralization; a smaller group of individuals (12% of right-handers and 15% of left-handers) showed weak lateralization; and a minority of around 7% (only left-handers) showed atypical right lateralization.

In line with this, our results (shown in figure 2) demonstrated a wide variability in laterality strength in left-handers, with a large overlap with the right-handed group. It is possible that what we observed as handedness differences in this study reflects the presence of a subset of 'atypical' participants, who are usually (but not always) left-handed [3,6,40]. Despite having doubled our sample size in this study, we would need more participants (particularly left-handers) to explore whether there are subsets of individuals *within* the handedness groups who show different patterns of lateralization. It may be the case that the laterality patterns shown in our SEM results follow the same groupings as those for lateralization strength. It may also be worth further investigating how lateralization varies as a function of the strength of manual preference. Two large fTCD studies have been explored this to date, both using a word generation task: the first (N = 326) [4] found a linear relationship, but the second (N = 310) [2] found no association. In addition, the relationship between handedness (as a continuous measure), language and gesture processing (praxis) was examined in an fMRI study (N = 94) by Häberling et al. [41]. With regard to language, they observed significant correlations between handedness and LIs for word generation in BA44/45, and for synonym decision in BA44/45 and the temporal lobe. However, using factor analysis across LIs from all praxis and language tasks, they found that handedness did not load significantly on the factor most strongly related to language laterality.

A final exploratory observation from this study concerns the interpretation of bilateral language. In left-handers, there were four tasks that did not show significant lateralization at the group level. We observed that the pattern of individual laterality within the left-handed group differed across tasks. For task B (Phonological Decision), individuals tended to show bilateral or weak left lateralization that was not consistent across sessions. However, for tasks C, E and F (Semantic Decision, Sentence

Comprehension and Syntactic Decision) individuals appeared to be split into different populations— some who were stably bilateral or stably right lateralized. This implies that the observation of bilaterality at the group level should be interpreted with caution: it may indicate that individuals form one stable distribution with weak laterality, or multiple subgroups that average out to show no lateralization at the group level.

One insight from this study is that tasks that do not appear lateralized at the group level may nevertheless show consistent individual differences in strength of laterality. The most striking example was the syntactic comprehension task, which had good test–retest reliability, but was not left-lateralized. Laterality indices from this task also showed significant correlations with other tasks that were left-lateralized (figure 3). This suggests that a focus on language laterality as a binary category may be misleading, as it overlooks stable individual differences in strength of lateralization that may manifest as left- or right-bias depending on the specific task.

A final recommendation that emerges from our study is that left-handers, who are often omitted from studies of cerebral lateralization, have the potential to provide a unique perspective on how language functions may dissociate. We may need larger-scale collaborative studies to recruit sufficiently large samples to get a clear picture of variation in this group, but our findings to date suggest this would be a worthwhile investment for advancing our understanding of how language is represented in the brain.

Ethics. All participants gave written, informed consent. Procedures were approved by the local ethics committee for either site: University of Oxford's Medical Sciences Interdivisional Research Ethics Committee or Bangor University School of Psychology Ethics Committee.

Data accessibility. All task materials, analysis scripts and anonymized data are available on Open Science Framework. The task stimuli, preregistration and data from the original study are available here: https://osf.io/tkpm2/ (doi:10.17605/OSF.IO/TKPM2). The data and analysis scripts for this paper are available here: https://osf.io/qetj5/ (doi:10.17605/OSF.IO/QETJ5).

Competing interests. We declare we have no competing interests.

Funding. This study was supported by the Wellcome Trust (082498) and by an Advanced Grant from the European Research Council (694189).

Acknowledgements. We would like to thank Dr David Carey for his helpful discussions and comments on this manuscript.

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
