## [Peer Review File · Royal Society Open Science]

Review History

RSOS-200696.R0 (Original submission)

Review form: Reviewer 1

Is the manuscript scientifically sound in its present form?

No

Are the interpretations and conclusions justified by the results?

Yes

Is the language acceptable?

Yes

Do you have any ethical concerns with this paper?

No

Have you any concerns about statistical analyses in this paper?

Yes

Recommendation?

Major revision is needed (please make suggestions in comments)

Comments to the Author(s)

This is an interesting and important update on a recent study investigating the multidimensionality of language. By significantly increasing the number of left-handed participants, the authors aimed to differentiate language lateralization patterns between left and right handers.

I was not a reviewer of the first paper, but I take the fTCD approach and experimental procedure was already thoroughly revised, so I will not go into that anymore. Instead, I will focus on some statistical clarifications and interpretation of the data.

Structural equation modelling (page 9): "Following our previous paper, we fixed the means and variances to be the same at each session -i.e. the parameter for the mean of task A at session 1 would be the same at session 2." Then, this is motivated by reasoning that a model with different means was no better than a model with fixed means given the good test-retest reliability of the measures. So, why not calculate the average LI for each measure in both sessions, thus creating a more reliable measure based on more observations? What does "fixing the means" mean? Making them equal? How can you make them equal when they are different measures. I'm probably missing something here, but I don't understand what.

The first paragraph on page 10 I also found difficult to grasp, but maybe that is because I'm not familiar with SEM.

The authors tested a one factor and two factor model. Given they used six different language tasks that could, at least theoretically, result in six factors, I was curious whether other models were tested as well (or why they stopped at two).

Thirty-seven more participants were included at Bangor of which 24 left handers and 10 right handers. The group of right handers now counted 43 participants. The authors mentioned in the introduction that the two factor model advantage of the first study appeared driven by a small amount of predominantly left handed individuals. And that for the majority of (right handed) participants the two factor evidence was weak. In the new study, with the extra 13 from Bangor, right handers favored the two factor model (albeit less strongly than the left handers). How much stronger was the evidence in right handers now compared to the first study? Was it substantially stronger or just a bit?

The authors are careful in their discussion not to over-interpret their findings. I particularly liked the suggestion that left handers may be a more heterogeneous group including people with weak or atypical laterality. We can only agree with the authors that the investigation of larger groups of left handers is warranted to provide clearer results. I'm just a little disappointed in the interpretation of the factors themselves. Although it is not the major aim of the study, it is not addressed in much detail here. In fact, the idea of two different language components is even a bit dismissed by advancing explanations based on modality or cognitive demand. When it comes to investigating multidimensionality of language lateralization, would the authors consider handedness differences as a relevant method to advance this field?

Minor comments:

Page 18 last sentence, shouldn't that read tasks B, C, E and F?

Page 22 middle "whether there whether there"

Review form: Reviewer 2

Is the manuscript scientifically sound in its present form?

Yes

Are the interpretations and conclusions justified by the results?

No

Is the language acceptable?

Yes

Do you have any ethical concerns with this paper?

No

Have you any concerns about statistical analyses in this paper?

Yes

Recommendation?

Major revision is needed (please make suggestions in comments)

Comments to the Author(s)

This manuscript addresses the question of whether there is more than one dimension (factor) underlying cerebral asymmetry for language, and whether this depends on handedness. The analysis is mathematically sophisticated, and I am not sure I fully understood all of it, but I do have a number of concerns:

1. It is not clear how handedness is defined. Some measures are based on performance (e.g., the peg-moving task) and some on preference (e.g., the Edinburgh Handedness Inventory), and these are not perfectly correlated. Both kinds of measure imply a continuum rather than a dichotomy, and there is always a subset who are ambidextrous rather than left- or right-handed. I am not really sure of the wisdom of simply dividing the sample into left- and right-handers, and carrying out parallel analyses – it's never a clean cut. An alternative approach is simply to use handedness as a variable and include it in a factor analysis, to see how and where it loads.
2. It has long been understood that laterality, including handedness itself, is more variable in left-handers (however defined) and early studies based on the effects of lateralized brain injury typically found no difference in asymmetry between left-handers and the ambidextrous, who tended to be lumped together for comparison with right-handers.
3. Earlier theories of handedness (Annett, McManus) proposed that left-handedness arose through lack of a genetic disposition to right-handedness, so that left-handedness was the outcome of a random process. This would include some right-handers. In its simplest, single-locus form, this theory probably no longer holds, but the idea that left-handedness (and some right-handedness) arises through chance rather than a genetic effect seems to have persisted in multi-locus models (e.g., McManus et al., 2013 – and see the predictions for language lateralization on p. 49 of that article). I'm wondering whether this, rather than some putative two-factor model, might explain the pattern of results (e.g., as shown in Figure 2). Perhaps there is simply an extra wash of randomness over the left-handers, creating more variance and bringing them closer to zero.
4. A two-factor model is always going to produce a better fit than a one-factor model. Perhaps, as the authors suggest, it is a better fit for left-handers than for right-handers, but I'm not sure what this could mean.
5. I found it hard to read meaning into the patterns of factor loadings. Did the authors consider rotating axes for a clearer picture of what the different factors might mean? This is fairly standard in factor analytic studies. Or perhaps try to rotate the two solutions into congruence with one another.

6. I'm not entirely clear how the factor analysis were done, but factor scores are only crudely estimated by multiplying original scores by factor loadings. In common factor space, where it is assumed that each variable (here, laterality index) has unique variance not present in the factor space, this problem has no exact solution. For an old article on this, see Horn (1965). Overall, I'm not sure that we gain from this study a coherent picture of the factor structure of cerebral asymmetry for language, or of exactly what the role of handedness might be.
Horn, J.L. (1965). An empirical comparison of methods for estimating factor scores. *Journal of Educational & Psychological Measurement*, 25, 313-322.

Review form: Reviewer 3 (Chris McManus)

Is the manuscript scientifically sound in its present form?

No

Are the interpretations and conclusions justified by the results?

No

Is the language acceptable?

Yes

Do you have any ethical concerns with this paper?

No

Have you any concerns about statistical analyses in this paper?

No

Recommendation?

Major revision is needed (please make suggestions in comments)

Comments to the Author(s)

I had difficulties accessing all of the data. Once the data are made available, perhaps with a rather clearer description of how the files should be run etc, then I will be able to provide a proper report. Please see the attached files for my preliminary report (Appendices A & B).

Decision letter (RSOS-200696.R0)

Dear Dr Woodhead,

The editors assigned to your paper ("Investigating multidimensionality of language lateralisation in left and right handed adults: an update on Woodhead et al. 2019") have now received comments from reviewers. We would like you to revise your paper in accordance with the referee and Associate Editor suggestions which can be found below (not including confidential reports to the Editor). Please note this decision does not guarantee eventual acceptance.

Please submit a copy of your revised paper before 18-Jul-2020. Please note that the revision deadline will expire at 00.00am on this date. If we do not hear from you within this time then it will be assumed that the paper has been withdrawn. In exceptional circumstances, extensions may be possible if agreed with the Editorial Office in advance. We do not allow multiple rounds of revision so we urge you to make every effort to fully address all of the comments at this stage. If deemed necessary by the Editors, your manuscript will be sent back to one or more of the original reviewers for assessment. If the original reviewers are not available, we may invite new reviewers.

- Data accessibility

<http://datadryad.org/submit?journalID=RSOS&manu=RSOS-200696>

- Competing interests

- Authors' contributions

All submissions, other than those with a single author, must include an Authors' Contributions section which individually lists the specific contribution of each author. The list of Authors should meet all of the following criteria; 1) substantial contributions to conception and design, or

acquisition of data, or analysis and interpretation of data; 2) drafting the article or revising it critically for important intellectual content; and 3) final approval of the version to be published.

- Acknowledgements

- Funding statement

on behalf of Dr Emma Hayiou-Thomas (Associate Editor) and Essi Viding (Subject Editor)
openscience@royalsociety.org

Associate Editor's comments (Dr Emma Hayiou-Thomas):

Associate Editor: 1

Comments to the Author:

One of the reviewers has requested additional information about accessing and analysing the data relating to this paper; they have submitted a preliminary report outlining what they would like to see before they go on to do a substantive review. We would like to progress the review process promptly, but also appreciate that preparing data can be quite time-consuming, so please let us know what a realistic time-frame would be.

Comments to Author:

Reviewers' Comments to Author:

Reviewer: 1

Comments to the Author(s)

This is an interesting and important update on a recent study investigating the multidimensionality of language. By significantly increasing the number of left-handed participants, the authors aimed to differentiate language lateralization patterns between left and right handers.

I was not a reviewer of the first paper, but I take the fTCD approach and experimental procedure was already thoroughly revised, so I will not go into that anymore. Instead, I will focus on some statistical clarifications and interpretation of the data.

Structural equation modelling (page 9): "Following our previous paper, we fixed the means and variances to be the same at each session –i.e. the parameter for the mean of task A at session 1 would be the same at session 2." Then, this is motivated by reasoning that a model with different means was no better than a model with fixed means given the good test-retest reliability of the measures. So, why not calculate the average LI for each measure in both sessions, thus creating a more reliable measure based on more observations? What does 'fixing the means' mean? Making them equal? How can you make them equal when they are different measures. I'm probably missing something here, but I don't understand what.

The first paragraph on page 10 I also found difficult to grasp, but maybe that is because I'm not familiar with SEM.

The authors tested a one factor and two factor model. Given they used six different language tasks that could, at least theoretically, result in six factors, I was curious whether other models were tested as well (or why they stopped at two).

Thirty-seven more participants were included at Bangor of which 24 left handers and 10 right handers. The group of right handers now counted 43 participants. The authors mentioned in the introduction that the two factor model advantage of the first study appeared driven by a small amount of predominantly left handed individuals. And that for the majority of (right handed) participants the two factor evidence was weak. In the new study, with the extra 13 from Bangor, right handers favored the two factor model (albeit less strongly than the left handers). How much stronger was the evidence in right handers now compared to the first study? Was it substantially stronger or just a bit?

The authors are careful in their discussion not to over-interpret their findings. I particularly liked the suggestion that left handers may be a more heterogeneous group including people with weak or atypical laterality. We can only agree with the authors that the investigation of larger groups of left handers is warranted to provide clearer results. I'm just a little disappointed in the interpretation of the factors themselves. Although it is not the major aim of the study, it is not addressed in much detail here. In fact, the idea of two different language components is even a bit dismissed by advancing explanations based on modality or cognitive demand. When it comes to investigating multidimensionality of language lateralization, would the authors consider handedness differences as a relevant method to advance this field?

Minor comments:

Page 18 last sentence, shouldn't that read tasks B, C, E and F?

Page 22 middle "whether there whether there"

Reviewer: 2

Comments to the Author(s)

This manuscript addresses the question of whether there is more than one dimension (factor) underlying cerebral asymmetry for language, and whether this depends on handedness. The analysis is mathematically sophisticated, and I am not sure I fully understood all of it, but I do have a number of concerns:

1. It is not clear how handedness is defined. Some measures are based on performance (e.g., the peg-moving task) and some on preference (e.g., the Edinburgh Handedness Inventory), and these are not perfectly correlated. Both kinds of measure imply a continuum rather than a dichotomy, and there is always a subset who are ambidextrous rather than left- or right-handed. I am not really sure of the wisdom of simply dividing the sample into left- and right-handers, and carrying out parallel analyses – it's never a clean cut. An alternative approach is simply to use handedness as a variable and include it in a factor analysis, to see how and where it loads.

2. It has long been understood that laterality, including handedness itself, is more variable in left-handers (however defined) and early studies based on the effects of lateralized brain injury typically found no difference in asymmetry between left-handers and the ambidextrous, who tended to be lumped together for comparison with right-handers.

3. Earlier theories of handedness (Annett, McManus) proposed that left-handedness arose through lack of a genetic disposition to right-handedness, so that left-handedness was the outcome of a random process. This would include some right-handers. In its simplest, single-locus form, this theory probably no longer holds, but the idea that left-handedness (and some right-handedness) arises through chance rather than a genetic effect seems to have persisted in multi-locus models (e.g., McManus et al., 2013—and see the predictions for language lateralization on p. 49 of that article). I'm wondering whether this, rather than some putative two-factor model, might explain the pattern of results (e.g., as shown in Figure 2). Perhaps there is simply an extra wash of randomness over the left-handers, creating more variance and bringing them closer to zero.

4. A two-factor model is always going to produce a better fit than a one-factor model. Perhaps, as the authors suggest, it is a better fit for left-handers than for right-handers, but I'm not sure what this could mean.

5. I found it hard to read meaning into the patterns of factor loadings. Did the authors consider rotating axes for a clearer picture of what the different factors might mean? This is fairly standard in factor analytic studies. Or perhaps try to rotate the two solutions into congruence with one another.

6. I'm not entirely clear how the factor analysis was done, but factor scores are only crudely estimated by multiplying original scores by factor loadings. In common factor space, where it is assumed that each variable (here, laterality index) has unique variance not present in the factor space, this problem has no exact solution. For an old article on this, see Horn (1965).

Overall, I'm not sure that we gain from this study a coherent picture of the factor structure of cerebral asymmetry for language, or of exactly what the role of handedness might be.

Horn, J.L. (1965). An empirical comparison of methods for estimating factor scores. *Journal of Educational & Psychological Measurement*, 25, 313-322.

Reviewer: 3

Comments to the Author(s)

I had difficulties accessing all of the data. Once the data are made available, perhaps with a rather clearer description of how the files should be run etc, then I will be able to provide a proper report. Please see the attached file for my preliminary report.

Author's Response to Decision Letter for (RSOS-200696.R0)

See Appendix C.

RSOS-200696.R1 (Revision)

Review form: Reviewer 1

Is the manuscript scientifically sound in its present form?

Yes

Are the interpretations and conclusions justified by the results?

Yes

Is the language acceptable?

Yes

Do you have any ethical concerns with this paper?

No

Have you any concerns about statistical analyses in this paper?

No

Recommendation?

Accept as is

Comments to the Author(s)

The authors have adequately addressed my suggestions. I have no further queries.

Review form: Reviewer 2

Is the manuscript scientifically sound in its present form?

No

Are the interpretations and conclusions justified by the results?

No

Is the language acceptable?

Yes

Do you have any ethical concerns with this paper?

No

Have you any concerns about statistical analyses in this paper?

Yes

Recommendation?

Reject

Comments to the Author(s)

In spite of the sophisticated statistical analysis I'm just not very convinced. Defining handedness by self report gives rather a crude dichotomy--handedness is really a continuum and I don't know what it really means to apply separate factor analyses to left and right handers. I see no real reason not to rotate axes to seek meaning in the factors, even though it's supposedly confirmatory rather than exploratory. For an alternative factor analytic approach which does treat handedness as a continuous variable, see Häberling, I.S., et al. (2016), *Cortex*, 82, 72-85. <http://dx.doi.org/10.1016/j.cortex.2016.06.003> 0010-9452, which should at least be cited. To my mind, at least, this provides a more interpretable solution, and I can't easily reconcile it with what is claimed here. It's still not clear to me that the interpretation is preferable to the Annett/McManus possibility that left handedness results from the absence of a right-handed influence; the randomness would

at the level of genotype, so one could still expect phenotypic correlations with left handedness. I'm also not sure the answers to the other reviewers are entirely convincing.

Review form: Reviewer 3 (Chris McManus)

Is the manuscript scientifically sound in its present form?

Yes

Are the interpretations and conclusions justified by the results?

Yes

Is the language acceptable?

Yes

Do you have any ethical concerns with this paper?

No

Have you any concerns about statistical analyses in this paper?

No

Recommendation?

Accept as is

Comments to the Author(s)

Many thanks to the authors for their detailed, interesting, and carefully thought through responses to my own comments and those of the other reviewers. The changes made to the paper have clarified a number of important issues, and of course one of the pleasures of open publication is that those readers who are really interested in the details of the issues can read the various comments and responses.

My only specific comment, which does not require any further changes, refers to the reply in 3.8 that, "this paper – already heavy going for those who are not familiar with SEM, would become indigestible with further analysis". I can see the point of this, but there is also an argument that if the phenomena require analyses which are heavy-going, then that is where the science has to go, as otherwise the phenomena will not be explored properly. Fortunately most physicists in their primary research papers do not restrict themselves to using only school-level maths or it seems unlikely that gravity-waves or whatever would have been discovered. At some point 'the reader' has to be willing to grapple with the issues, even if they are difficult. End of sermon!

I will end by reiterating where I started reviewing this study, that I think this is one of the most interesting studies on laterality to emerge in a number of years, and the authors are to be congratulated on it, and it will provide provocative food for thought for a number of years.

Decision letter (RSOS-200696.R1)

Dear Dr Woodhead

On behalf of the Editors, we are pleased to inform you that your Manuscript RSOS-200696.R1 "Investigating multidimensionality of language lateralisation in left and right handed adults: an update on Woodhead et al. 2019" has been accepted for publication in Royal Society Open Science subject to minor revision in accordance with the referees' reports. Please find the referees' comments along with any feedback from the Editors below my signature.

Please submit your revised manuscript and required files (see below) no later than 7 days from today's (ie 04-Dec-2020) date. Note: the ScholarOne system will 'lock' if submission of the revision is attempted 7 or more days after the deadline. If you do not think you will be able to meet this deadline please contact the editorial office immediately.

on behalf of Dr Emma Hayiou-Thomas (Associate Editor) and Essi Viding (Subject Editor)
openscience@royalsociety.org

Associate Editor Comments to Author (Dr Emma Hayiou-Thomas):

Comments to the Author:

Thank you for such a thorough and thoughtful response to the reviews. The additional wording clarifying SEM modelling and the (speculative) interpretation of the factors is particularly helpful. I also like the new title! One final point which I think requires acknowledgement in the discussion is the issue of the dimensionality of handedness, which is a point raised by two of the reviewers. It's now clear in the ms that you used a dichotomous self-report measure, but as you say in your response to the reviews, a continuous measure of handedness would be an alternative approach, and one which would be interesting to take in future work.

Reviewer comments to Author:

Reviewer: 2

Comments to the Author(s)

In spite of the sophisticated statistical analysis I'm just not very convinced. Defining handedness by self report gives rather a crude dichotomy--handedness is really a continuum and I don't know what it really means to apply separate factor analyses to left and right handers. I see no real reason not to rotate axes to seek meaning in the factors, even though it's supposedly confirmatory rather than exploratory. For an alternative factor analytic approach which does treat handedness as a continuous variable, see Häberling, I.S., et al. (2016), *Cortex*, 82, 72-85. <http://dx.doi.org/10.1016/j.cortex.2016.06.003> 0010-9452, which should at least be cited. To my mind, at least, this provides a more interpretable solution, and I can't easily reconcile it with what is claimed here. It's still not clear to me that the interpretation is preferable to the Annett/McManus possibility that left handedness results from the absence of a right-handed influence; the randomness would at the level of genotype, so one could still expect phenotypic correlations with left handedness. I'm also not sure the answers to the other reviewers are entirely convincing.

Reviewer: 1

Comments to the Author(s)

The authors have adequately addressed my suggestions. I have no further queries.

Reviewer: 3

Comments to the Author(s)

Many thanks to the authors for their detailed, interesting, and carefully thought through responses to my own comments and those of the other reviewers. The changes made to the paper have clarified a number of important issues, and of course one of the pleasures of open publication is that those readers who are really interested in the details of the issues can read the various comments and responses.

My only specific comment, which does not require any further changes, refers to the reply in 3.8 that, "this paper – already heavy going for those who are not familiar with SEM, would become indigestible with further analysis". I can see the point of this, but there is also an argument that if the phenomena require analyses which are heavy-going, then that is where the science has to go, as otherwise the phenomena will not be explored properly. Fortunately most physicists in their primary research papers do not restrict themselves to using only school-level maths or it seems unlikely that gravity-waves or whatever would have been discovered. At some point 'the reader' has to be willing to grapple with the issues, even if they are difficult. End of sermon!

I will end by reiterating where I started reviewing this study, that I think this is one of the most interesting studies on laterality to emerge in a number of years, and the authors are to be congratulated on it, and it will provide provocative food for thought for a number of years.

===PREPARING YOUR MANUSCRIPT===

===PREPARING YOUR REVISION IN SCHOLARONE===

- If you are providing image files for potential cover images, please upload these at this step, and inform the editorial office you have done so. You must hold the copyright to any image provided.
- A copy of your point-by-point response to referees and Editors. This will expedite the preparation of your proof.

- Ensure that your data access statement meets the requirements at <https://royalsociety.org/journals/authors/author-guidelines/#data>. You should ensure that you cite the dataset in your reference list. If you have deposited data etc in the Dryad repository, please only include the 'For publication' link at this stage. You should remove the 'For review' link.
- If you are requesting an article processing charge waiver, you must select the relevant waiver option (if requesting a discretionary waiver, the form should have been uploaded at Step 3 'File upload' above).
- If you have uploaded ESM files, please ensure you follow the guidance at <https://royalsociety.org/journals/authors/author-guidelines/#supplementary-material> to include a suitable title and informative caption. An example of appropriate titling and captioning may be found at https://figshare.com/articles/Table_S2_from_Is_there_a_trade-off_between_peak_performance_and_performance_breadth_across_temperatures_for_aerobic_scope_in_teleost_fishes_/3843624.

Author's Response to Decision Letter for (RSOS-200696.R1)

See Appendix D.

Decision letter (RSOS-200696.R2)

Dear Dr Woodhead,

Happy new year!

It is a pleasure to accept your manuscript entitled "An updated investigation of the multidimensional structure of language lateralisation in left and right handed adults" in its current form for publication in Royal Society Open Science.

on behalf of Dr Emma Hayiou-Thomas (Associate Editor) and Essi Viding (Subject Editor)
openscience@royalsociety.org

Appendix A

Report: Investigating multidimensionality of language lateralisation in left and right handed adults: an update on Woodhead et al. 2019 (RSOS-200696)

This study contains the second most exciting dataset on variation in human cerebral lateralisation that I have ever seen, and I have seen both within the past year. The most exciting dataset was that described by Emma Karlsson in her PhD Thesis (Bangor, 2019), supervised by David Carey, which described a systematic search for a large number of left-handers and individuals with atypical language dominance, as well as some control right-handers, and then used fMRI to assess their lateralisation on a range of left and right-hemisphere tasks. To say the least I was surprised that it was not mentioned in the current study given that Emma is a co-author here. I couldn't help wondering whether that might have reflected an analytical and methodological difference between what I will call the Karlsson fMRI study and the present "Oxford" fTCD study (actually Oxford/Bangor), with the former looking primarily at differences in direction of lateralisation and the latter at differences in degree of lateralisation. That can be returned to later.

The present study is actually two, with the 2019 Oxford data of Woodhead et al, and the present (2020) Oxford+Bangor data of Woodhead et al. For convenience I will describe them as the 2019 and 2020 studies. The basic protocol of the 2019 study was good, with six carefully selected tasks, each repeated in a proper test-retest protocol (i.e. with a delay of several days or more). That gives a powerful set of data to analyse.

The 2019 study was pre-registered, although I have to confess to not always being a fan of pre-registration when studies are in novel territory and exploration is much more the order of the day; of course the medic in me sees the need for pre-registration in clinical trials and the like, but unlike in situations such as that which are 'ordinary science' (to use Kuhn's terms), when data instead are exploratory and one is moving into the territory of 'extra-ordinary science', pre-registration seems to be like tying one or both hands behind one's back. I will therefore ignore the pre-registering aspect here. When someone replicates the Karlsson or Oxford studies then that will be the time for pre-registration.

A clear limitation of the 2019 study was the small number of left-handers, and a strong sense that they were behaving differently to the right-handers, with insufficient numbers to address the issues properly. The 2020 study takes very seriously the whole idea of handedness being important for studying language lateralisation (although perhaps that should not come as a surprise given many results ever since the time of Broca). An attempt is made to get a study balanced for handedness, with most of the left-handers being brought in from Bangor. At that point the data get really interesting.

The present Oxford study is also interesting because of the care and elegance of the design – six related but different language tasks, each with proper test-retest reliability assessed on separate occasions. The psychometrics is also good, and it is a pleasure to see structural equation modelling, which as the authors rightly say, is pretty unusual in laterality research. That may be with good reason, but that can be returned to later.

The data are also open, and while there were some hiccoughs on the first iteration of trying to get the data from osf.io, the authors responded very well to an earlier, preliminary report, and I have now successfully rerun all of the analyses, the improvements mostly being due to the use of the `osfr()` functions in *R* which simplified everything from a user's point of view. The 'readme' file was

also extremely helpful. I even managed to get the pirate plots running, although I could only agree with the comment in the code of “# Holy moly that was a lot of work for one plot”... In general the code was exemplary in its clarity, and well written (and I would not like my own code exposed to public gaze in this way). Working through the 2000+ lines of code was fascinating, and I learned a lot and understood even more about the subtleties of the data analysis. I suspect the programs would make a good teaching file for explaining excellence in coding to postgrads, both for how R works and how it can be presented in a clear way. That is a hidden benefit of the open source movement which is not always mentioned. Of course it also means that others can analyse and re-analyse the data themselves, see below, and can also use the code themselves (plagiarism not being a sin but a compliment in the open data/code world).

That is enough by way of broad introduction. It should be obvious that I think the study is excellent in many ways, and that it needs publication. That doesn't mean it is perfect, nevertheless, and some comments and possible changes might help it.

More specific comments:

1. **Handedness.** The paper is about handedness. However, nowhere can I see any statement or discussion about how it was measured. Edinburgh? The Bishop peg moving task? Or perhaps merely self-description? I suspect the latter or we would have been told more. Either way, we need to be told something. That is particularly the case when on p.22 when the paper says, “It may also be worth further investigating how lateralisation varies as function of the strength of manual preference”; so did the present study not have any measure of strength of manual preference? [Note added later: While browsing through the data I spotted that particdat_Bangor has a column called EHI, but there is no equivalent in the A2 data].
2. **How were the left-handers recruited?** Again very little on finding some relatively unusual individuals. I may be putting 2+2 to make 5, but I did wonder if some of these left-handers were those previously used by Emma Karlsson in her thesis. Again, a clear statement on recruitment would be useful. Actually I would rather like it if they were the same individuals as then there would be both fTCD and fMRI data in the same individuals, which is precious rare.
3. **Removal of trials, cases, etc..** At times I did wonder if too many participants and/or observations were being discarded. I can fully accept technical errors, spikes, etc.. Much more concerning for me was that there seemed an obsessive, almost fetishistic, determination to remove outliers, with everything trying to be forced into normal distributions. The normal distribution is convenient, but it is not always the road to analytic salvation. Sometimes outliers are there because they are different and meaningful. That single left-hander in the 2019 study was the bellwether for other possible problems, but it was an outlier. If one is interested in variation, then removing individuals who are too variable is not a good analytic strategy. If it were height that were being analysed then removing outliers means one would never discover either achondroplasia or acromegaly in the population... I did wonder about an analysis/ dataset with as little removed as possible (technical issues aside).
4. **Normality.** From the point of view of the replication crisis, the removal of participants because they do not fit some pre-decided analytic model, particularly to do with normal distributions, is perhaps a relative of ‘harking’, with theoretically relevant decisions being made before the data are analysed (good), but the data being pre-constrained into a mould which is perhaps not appropriate (bad), only explaining some of the data and discarding awkward or difficult cases. In statistical terms, there are good reasons to believe that

laterality data are not normally distributed, but probably are mixtures in some way of normal distributions, perhaps symmetrically arranged around zero (see McManus, 1983, *Cortex*, 19: 198-214).

5. **Estimating reliability of laterality measures.** One of the several joys of this and the 2019 study is to see a serious attention to reliability. At this point it is worth saying that there are three traditional approaches to reliability. Internal consistency (e.g. alpha), test-retest reliability/stability, and stability/equivalence (different versions of a test on different occasions). The latter is not available as yet (presumably fTCD and fMRI on different occasions), but this study provides test-retest reliability. I'm not sure it is labelled as such but it can be seen in figure 3, separately for R and L handers, as the correlation of A1-A2, B1-B2 etc.. As the paper says, the reliabilities are not good for A. The test-retest correlations are all higher for left-handers than right-handers (e.g. .8 vs .52 for B1-B2) but that presumably reflects the greater variance in the left-handers scores (visible in figure 4). It may well be that the variance would be greater if 'outliers' had not been removed quite so extensively.
6. **Internal consistency.** Looking into the code shows that as well as test-retest reliability there are also odd-even measures of internal consistency, only however for the analyses using the 'peak difference' method rather than the 'mean difference' method. I only looked at the peak difference odd-even consistency measures but they are substantially higher than test-retest reliabilities (and I presume the same would be the case for mean difference measures). That is interesting, as I think in previous studies the odd-even measures have been reported as reliability estimates for data similar to these. What is really needed is a more sophisticated structural model which contains odd-even scores nested within test and retest. That should give a better sense of the true reliability of measures such as these, and also allow comparison with previous studies. An interesting question is whether participants change their lateralisation from occasion to occasion, perhaps due to strategy differences.
7. **Mixtures.** Mixtures and mixture modelling are mentioned briefly in relation to the Mazoyer et al 2014 study, but never again appear. "It is possible that what we observed as handedness differences in this study reflects the presence of a subset of these 'atypical' participants in the left handed group but not the right handed group." That opens a whole can of worms, not least about the earlier comment on removing all points which are 'outliers'. Of course the phrase, "in the left handed group but not the right handed group" is also problematic, as in the Karlsson Bangor fMRI study it is very clear that there are atypical right-handers, so excluding them here from being atypical does not seem likely. There are also many right-handed patients in the literature with 'crossed aphasia', even if they are proportionately rare. The Carey and Johnstone meta-analysis (your ref 5) makes that clear.
8. **The nature of the measurement scale in laterality studies.** This is perhaps my major theoretical criticism of the present study, and it eventually comes back to a) mixture distributions and b) the different approaches of the Karlsson and Oxford studies. To start with though we need to get to grips with what is being measured and what laterality is all about.
 - a. Handedness, as is clear to anyone, particularly in EHI scores, is not normally distributed. Other lateralities are also clearly not normally distributed – consider the side of the heart in the chest, which is mostly varies in distance to the left of the midline, but in situs inversus varies in distance to the right of the midline. The distribution is bimodal, with the modes placed symmetrically around zero. Something similar applies to the EHI, although the mode of the left-handers is closer to zero than the mode of the right-handers, and the data are clearly censored at the

- extremes. Skills such as the Tapley-Bryden task are much more obviously bimodal normal (Laterality, 2016, 21: 371-396) with symmetric modes.
- b. The present study though considers a continuum of scores, explicitly assumed to be normally distributed. The dependent variable is the average left-sided blood flow minus the average right-sided blood flow, with zero indicating no difference. Most but not all of the differences are positive, with some being negative. Although that is what the data actually are, there are few places in the analyses where zero plays any special role. They are there in figure 2, but I had to draw them in for figure 4. The SEM analyses though simply know nothing about zero and the fact that >0 means “on the left” and <0 means “on the right”. It is worth remembering that Broca didn’t say that language on average was on the left, but in individuals he said it was either in the left or in the right (just as the heart is not on average on the left side in any meaningful sense). To emphasise the point, if the data were, say, reaction times, then the models would all look identical to their present form. So where does lateralisation really come into these models?
 - c. The answer to the above question is, I think, that it doesn’t. But that is to miss out on the key aspect of lateralisation – that it is lateralised...! Here I think is the key way in which the Karlsson and Oxford approaches are so different. Oxford treats lateralisation as a normally distributed continuum, whereas Karlsson treats it ultimately as binary (and for fMRI data that is seen in her very important table 3.11 where for five tasks, 24 out of the 32 possible combinations are found; and for the 64 possible combinations with handedness included, 30 combinations were found, and that was based on just 67 participants). Comparing right and left-handers, of 24 right-handers, 13 (54%) had no atypical lateralities and 1 (4%) had all five as atypical, whereas in 43 left-handers, only 4 (9%) had all five lateralities as typical, and 3 (7%) had all five lateralities as atypical. The analyses of the current study can’t really get at such phenomena at all, although it is possible to ask whether mean scores are significantly different from zero, but little is done with those differences.
 - d. Immediately the question is raised as to whether the Karlsson and Oxford approaches are incompatible, or instead are talking about similar things from entirely separate perspectives. If laterality can be considered as a bimodal mixture distribution (back to that Cortex paper) then there are separate questions concerning what we can call direction of lateralisation and degree of lateralisation. Karlsson is concerned with direction and Oxford with degree. Both are valid, but they answer different questions in different ways. Simply analysing overall scores also confounds direction and degree and can be very difficult to interpret (see the toy example in figure 1 of McManus 1983). The SEM modelling here is unable to separate direction and degree.
 - e. It is possible to look at the present data in a manner similar to that of Karlsson, simply by dichotomising the LI score for each task at zero. Very interestingly the code for the statistical analysis, at lines 1273 to 1284, asks “Does the number of left lateralised tasks per participant vary with handedness?”. The answer, just as with Karlsson’s data, is a resounding, Yes. Overall, of 74 participants, only 24 had all six language tasks left lateralised; and indeed 4 had none of them left-lateralised, the other 46 having 1 to 5 not left-lateralised. Some of that may be measurement error, but it is unlikely all to be so. In relation to handedness, $20/43= 46.5\%$ right-handers and $4/31= 12.9\%$ left-handers had all six tasks left-lateralised. The Karlsson and Oxford studies are very different in many ways, but the pattern of differences in

direction of lateralisation is broadly very similar. I've played around with matrices of tetrachoric correlations from the two studies and there is a lot of similarity in the Karlsson fMRI data and the Oxford fTCD data.

- f. What is it that SEM can't do that analysing direction of laterality can? Essentially the difference is that SEM treats data as being on a continuum, but it has no sense that zero has any substantive meaning at all, beyond being a value between +1 and -1. It is as if one were measuring temperature and ignored the fact that 0 has an important meaning on the Centigrade scale. For analysing air temperature or the length of railway tracks that might not matter very much, but for water there is a key change at zero (which is why zero is set there), and there is a qualitative difference between +1 (liquid water) and -1 (solid ice). There is what physicists call a phase change at 0, and everything is different on one side of 0 and the other. Probably cerebral lateralisation can be construed in the same way, with a phase shift at L-R differences of zero. Any analysis of lateralisation needs to recognise that.
 - g. The answer is probably to fit mixture distributions within SEM, which could handle the sort of mixture distributions and phase changes suggested above, but I wouldn't recommend it for the present study.
9. What are the implications of the previous section for this manuscript? I don't think that the paper needs more analyses which take direction and degree into account (and that will provide plenty of material for future work). However the paper does need to discuss differences in direction and degree. That is really necessary because of the seemingly very different approaches of the Karlsson and Oxford groups, and the present paper, not least because of Emma being an author, needs some form of consideration of where the studies are similar and where they are different. Also needed is to recognise the limitations of the current SEM modelling for handling issues of direction and degree. Taking that further can await a later paper.
- 10. Left hemisphere functions?** The entire paper is about 'language lateralisation' (title) and the implicit assumption is that all of the tasks are left lateralised. That is a bit hard to reconcile however with the overall means, which for SemDec are small and not significantly different from zero, and particularly for Jabber, where the overall mean is clearly negative (-.24), with se of .18, and 40/74 (54%) participants right-lateralised. Are these bilateral tasks, in some sense, or do they have right-hemisphere components to them? Needless to say it would be interesting to know what they looked like with fMRI. I am not sure there is any comment in the paper on these particular measures, but it is perhaps needed. They do correlate with the other measures, but I am not sure that clarifies things. If Knecht et al had started with a Jabberwocky task they might well have abandoned fTCD... At that point I did wish that Knecht et al's word generation task had been included, particularly as it has been called, "the gold standard task for assessing language lateralisation" (Laterality, 2012, 17: 694-710).
- 11. Finally, the title.** Is it just me, or does the title utterly underplay the importance and interest of this study? It sounds like a mere scholarly footnote, perhaps adding a few more cases, instead of addressing the key issue in the 2019 study which was the lack of left-handers in reasonable numbers. Maybe something like, "The multidimensional structure of language lateralisation: An fTCD study of six language tasks on two occasions in 31 left-handed and 43 right-handed adults". You get the idea.

Minor comments

12. p.5. "In order to mitigate the potential confound of testing site, we also aimed to test at least 10 right handed participants at the new site (Bangor)". Are there comparisons of the

Oxford and Bangor data? If so, I couldn't find them. And is 10 right-handers in Bangor really sufficient?

13. Handling of missing data in the SEM. Looking carefully at the code, I think it is the case that missing values were handled by means of FIML, full-information maximum likelihood. That is good, but I am not sure it is ever explained in the text. For those less experienced in SEM, there may well be confusion as figure 3 shows correlation matrices, but these presumably have different Ns in the various cells. Much basic SEM, pre-FIML, uses covariance or correlation matrices, and it was tempting at first to assume that you had analysed the matrices presented in figure 3. Analysing correlation matrices would have been problematic since the measures are on the same measurement scale (mean L-R difference in blood flow) and therefore it would have been appropriate here to use covariance matrices rather than correlation matrices, which have all sorts of inherent problems. I can see the expository value of the correlation matrices, but clarification is needed of what you actually did. There may even be an argument for presenting covariance matrices and just accepting it.
14. You are using SEM models where means are estimated. However the diagram in figure 1 does not reflect that. The diagrams in figure 4 of the 2019 paper make it clearer that both means and variances are being analysed, although I confess that I find it less than clear when means are presented in one diagram and variances in another. Typically both are presented in the same diagram (see the introductory material on OpenMx in <https://openmx.ssri.psu.edu/sites/default/files/OpenMx%20beta%20User%20Guide.pdf> where, say, section 2.2.2 on p.65 presents a clear model with both means and variances). For the present case where there are more variables the solution is probably to have just three variables [A1, B1,...F1] and keep the model easier to read.
15. **Model Fit Statistics.** Model Fit Statistics in the 2020 and 2019 papers are presented very differently. Degrees of freedom are complicated, and are likely to confuse the beginner (and it is suggested that most of those interested in laterality are beginners as it is so rare). DF are easiest to understand when modelling covariance matrices. A 12x12 covariance matrix similar to that in figure 3, has 12 diagonal elements and 66 off-diagonal elements, making 78 df. If means are included there are another 12 elements, making 90 df in total. Models then consume various of those dfs, leaving the others in a residual, which in simple modelling is the df of the chisquare goodness of fit statistic. At that point, it becomes confusing in the 2019 paper when the table says there are 409 df for the fully saturated model (because of FIML). Various of those df do get eaten up by parameters. Constraining some of the parameters, as with the task effect model then gives another 12 df and so on. None of that is obvious. Less obvious still is that table 4 in the 2020 paper doesn't mention df, but instead describes parameters, and gives delta df. That is clearer. The table though has AIC and not BIC as in 2019, the CFIs look nothing like the 2020 values, and there are chi-square statistics with no dfs. The 2020 table has no chisquare statistics or associated p values. This doesn't feel like an 'update' and it also feels rather confusing even when one has used SEM for many years. Some consistency and explanation would probably help those not used to SEM.

Summary. This is an exciting and interesting paper which opens up many questions concerning the nature of lateralisation across tasks, and between people. Its inclusion of a substantial number of left-handers, and its careful attention to the details of the design and the analysis mean it can be used to explore many aspects of lateralisation.

Chris McManus

University College London

Appendix B

Preliminary report on data and code for RSOS-200696, “Investigating multidimensionality of language lateralisation in left and right handed adults: an update on Woodhead et al. 2019”

This report entirely concerns the data and the code for this paper, and the lack of any raw Doppler data for the present study, coupled with a host of minor problems in processing the 2019 data, needs resolving before a proper report can be written. Before those pedantic issues, let me start by giving the opening paragraph of the full report, which I have started to write:

“This study contains the second most exciting dataset on variation in human cerebral lateralisation that I have ever seen, and I have seen both within the past year. The most exciting dataset was that described by Emma Karlsson in her PhD Thesis (Bangor, 2019), supervised by David Carey, which described a systematic search for a large number of left-handers and individuals with atypical language dominance, and then used fMRI to assess their lateralisation on a range of left and right-hemisphere tasks. To say the least I was surprised that it was not mentioned in the current study given that Emma is a co-author. I couldn’t help wondering also whether that reflected an analytical and methodological difference between the Bangor study and the Oxford study, with the former looking primarily at differences in direction of lateralisation and the latter at difference in degree of lateralisation. That can be returned to later.”

Given my enthusiasm I was clearly eager to look in detail at the data and to explore them as a part of writing a proper report. That is where the problems started...

The data and the code. The distinguishing feature of open access is that “Datasets and code should be deposited in an appropriate, recognised, publicly available repository”. The journal’s instructions to reviewers suggest that a reviewer’s job is to ensure that is possible. That therefore means checking that the data are available and that the code works. At best that is tedious and at worst time-consuming. In the present case it is made harder by the present (2020) paper being an extension of the 2019 paper which has already been published, so that there were two separate sets of data which were being combined.

Although I began by checking out the data in 2020 paper, I immediately discovered that also meant checking out the 2019 paper, for some data and analyses are only there. Either the reviewers for the 2019 paper were more fluent than I am at navigating *osf.io*, or they are better at getting other people’s R programs to run. The present programs are, it must be said, well written, with clear commenting etc, but understanding other people’s code is rarely straightforward. The main problem is getting the data in and out.

The 2019 study.

- Data are in <https://osf.io/tkpm2/> according to Woodhead et al 2019.
- Starting there I found a file called A2_doppler-analysis_v2.r which I downloaded.
- Inevitably that program did not run on my computer because directory structures etc are wrong. However there is no list of files/directory structures which are needed for input and output (and for instance I finally got it running today when I found I was missing an output subdirectory in line 623 which had the unhelpful error message, “Error in file(file, ifelse(append, "a", "w")) : cannot open the connection”. By then I had downloaded the file called A2_fTCD_Data.zip, but still had several other files to find in different locations (e.g. triallist.xlsx).
- The missing output directory was the last in a long chain of problems, not helped by there being multiple websites at *osf.io*. I thought the Wiki might help but it didn’t.
- So what did I want?
 - A helpful file called something like “aaareadme.txt”, as all packages used to have once upon a time which would get me going on running the software. A step by step guide for idiots in other words (or at least people who haven’t been working with these data for several years).
 - That link would guide me to a single zip file which contained the R program and all the necessary input and output files and directories.

- I can then put all of the zipped files in some one convenient place on my computer. The R program would then say that all I needed to do was to change a single directory name at line xyz, so that `setwd()` would immediately find everything I needed. That zip file could also include a dummy file in the output directory.
- So why wasn't that there? My guess is because i) no-one else had ever tried to get all of this running from scratch given the material at osf.io, (and that includes perhaps other reviewers – the 2019 paper is after all published), and ii) because those who develop and use something on a daily basis never quite realise how difficult things can be for those coming afresh (and at least I am pretty fluent in R). I'm tempted to say that the journal perhaps should be doing all of this, not a hapless reviewer, who has spent too many hours already (including writing what is only a preliminary report).
- While we are on the 2019 paper:
 - The original protocol. The website does seem to have the amended protocol but only after endless searching did I find something at <https://osf.io/f7m8n/>. That doesn't seem to be mentioned in the 2019 paper. The search function at osf.io is not very friendly either.
 - I was slightly confused by <https://osf.io/tkpm2/> also having files referring to Bangor, which is presumably the 2020 paper, and therefore is in the future as far as the 2019 paper is concerned.

The 2020 study.

- The data etc are at <https://osf.io/qetj5/> according to the 2020 paper.
- There was a file called README.md, but sadly it was only two lines long and not very informative.
- Once again I started with the raw Doppler data (and I was particularly interested in them since the present study uses the mean difference rather than the peak difference method of Deppe et al). The file `A2_1_doppler_analysis_Emma.R` was clearly the place to start.
- Running the program showed that it was looking for files in a place such as `"/Users/zoe/Dropbox/Bangor_fTCD/Emma_data/142/1/"` with a name such as `A2_142_A1.exp`. However there seems to be no sign of those .exp files in <https://osf.io/qetj5/>. At that point the analysis of the Doppler data stopped.
- Whether I have missed these data or if they are hiding in plain sight I have no idea. Either way, it would be helpful to have a rather better guide as to where the data are to be found.

I did carry on from there with both sets of data, and think I have managed to read the combined summary data from the 2019 and 2020 papers and looked at other aspects of the study.

On a positive note, and despite the time involved, I have to say I have learned a lot by having been able to look through the code line by line and see how it has been processed. I do not doubt therefore the benefits of having open data and open code, but it could perhaps be made a little easier for those on the receiving end of it. As mentioned earlier, this may perhaps be something that the journal needs to consider – perhaps employ a computer-literate psychology PhD student to see how well they can get the data running, and perhaps at the same time write a 'how to do it' account for the next user. I think this is now my fourth paper with open data, and it has not been straightforward in any of them – but as I have said, I have learned a lot from each of them.

Appendix C

Reviewer: 1

This is an interesting and important update on a recent study investigating the multidimensionality of language. By significantly increasing the number of left-handed participants, the authors aimed to differentiate language lateralization patterns between left and right handers. I was not a reviewer of the first paper, but I take the fTCD approach and experimental procedure was already thoroughly revised, so I will not go into that anymore. Instead, I will focus on some statistical clarifications and interpretation of the data.

1.1: Structural equation modelling (page 9): “Following our previous paper, we fixed the means and variances to be the same at each session –i.e. the parameter for the mean of task A at session 1 would be the same at session 2.” Then, this is motivated by reasoning that a model with different means was no better than a model with fixed means given the good test-retest reliability of the measures. So, why not calculate the average LI for each measure in both sessions, thus creating a more reliable measure based on more observations? What does “fixing the means” mean? Making them equal? How can you make them equal when they are different measures. I’m probably missing something here, but I don’t understand what.

We have reworded (as shown below) to clarify that with SEM we are fitting a formal model to observed data. When we talk of 'fixing the means' this indicates that our model uses the same estimate for the mean for a given measure for both test occasions. Imposing the constraints on the measures at different time points effectively does what the reviewer suggests but does not reduce the number of observations used in the model. Using the reviewer's suggestion would reduce power and would not be advisable given that our current sample size is fairly modest already. In our prior report with a subset of these data (Woodhead et al 2018) we explicitly tested this aspect of our model.

“The analysis works by comparing the covariance matrix observed from the data with the covariance matrix implied by the structural model. Good fit is obtained when the observed pattern agrees well with model predictions.” (p.9)

“...we fixed the means and variances to be the same at each session – i.e. the parameter for the mean of task A at session 1 was set to be the same at session 2.” (p.10)

1.2: The first paragraph on page 10 I also found difficult to grasp, but maybe that is because I’m not familiar with SEM.

We have improved the description of factors on p.10 and added a reference to a basic SEM textbook for readers unfamiliar with SEM.

“The ovals in Figure 1 are factors, or latent variables. In SEM, latent variables are not observed directly, but estimated from the observed variables (also known as the indicators). A well-known example is that ‘g’ is a latent variable that is a representation of intelligence as measured in multiple different tasks (indicators). The factor structures determine the covariance between the indicators. The models tested had either one or two factors. To

ensure identifiability of the models when estimating parameters, we follow the usual convention of fixing certain parameters (Kline, 2011).”

1.3: The authors tested a one factor and two factor model. Given they used six different language tasks that could, at least theoretically, result in six factors, I was curious whether other models were tested as well (or why they stopped at two).

In confirmatory factor analysis, the purpose is to find structure and shared variance in explained an underlying and unobserved measure with a predefined structure. We only tested specific hypotheses regarding a one factor and two factor solution; this was motivated by our previous study where we obtained a good model fit with 2 factors. The goal in SEM is essentially data reduction, finding the most parsimonious model compatible with the data, so one would not start with a model with many factors. In addition, if you add factors, then the sample size required for adequate power increases. Formally, a model with 6 factors would suffer from what are termed 'lack of identifiability', which occurs when the number of parameters to estimate is greater than the number of observations.

1.4: Thirty-seven more participants were included at Bangor of which 24 left handers and 10 right handers. The group of right handers now counted 43 participants. The authors mentioned in the introduction that the two factor model advantage of the first study appeared driven by a small amount of predominantly left handed individuals. And that for the majority of (right handed) participants the two factor evidence was weak. In the new study, with the extra 13 from Bangor, right handers favored the two factor model (albeit less strongly than the left handers). How much stronger was the evidence in right handers now compared to the first study? Was it substantially stronger or just a bit?

It is not possible for us to quantify the difference in evidence strength: models with different samples cannot be compared directly, only nested models (i.e. different model structures but using exactly the same data). The best indication of the difference between left and right handers is that for left handers the leave-one-out validation supported the 2 factor model at $p < .001$ in every iteration; whereas, for right handers, the validation supported the 2 factor model in only 27 out of 43 iterations, with generally weak p -values.

We have added a note to the Discussion to clarify this issue:

“Note that using the SEM approach it is not possible to directly compare the strength of evidence for the two factor model between left and right handers: model comparison is only valid for nested models, where the data remains the same but the model changes.” (p. 22)

1.5: The authors are careful in their discussion not to over-interpret their findings. I particularly liked the suggestion that left handers may be a more heterogeneous group including people with weak or atypical laterality. We can only agree with the authors that the investigation of larger groups of left handers is warranted to provide clearer results. I’m just a little disappointed in the interpretation of the factors themselves. Although it is not the major aim of the study, it is not addressed in much detail here. In fact, the idea of two different language components is even a bit dismissed by advancing explanations based on modality or cognitive demand. When it comes to

investigating multidimensionality of language lateralization, would the authors consider handedness differences as a relevant method to advance this field?

We agree that this dataset does not conclusively tell us the key characteristics of the two factors, but it does suggest some hypotheses to be tested in future studies. We have speculated further and now state:

“The biggest differences in the factor loadings between groups (as shown in Figure 5) were observed in the Semantic Comprehension and Syntactic Decision tasks (E and F). Both of these tasks required receptive sentence level comprehension. Hence, a speculative hypothesis for future work is that in left handers (or a subset of individuals, the majority of whom are left handed), the language processes involved in phonology, semantics and expressive speech co-lateralise together, but sentence comprehension can lateralise independently. The critical language process in tasks E and F that drove this second factor may have been speech perception (as both used auditory stimuli) or it may have been related to sentence level semantics or syntax. One quite consistent finding from the literature is that speech production is more strongly lateralised than comprehension (29–34) c.f. (35), but this does not necessarily imply that those two functions lateralise independently – despite having different means, they may still be correlated. In our previous paper, we motivated task selection using Hickok and Poeppel’s dual stream model of speech perception (36), which proposes a division of labour between dorsal areas involved in sensorimotor processing of speech, and ventral areas involved in lexical access and comprehension. However, although it might be tempting to identify Factor 1 with the dorsal stream, and Factor 2 with the ventral stream, that would not account for the loading of the semantic decision task on Factor 1. A more parsimonious explanation might be that processes relating to sentence level comprehension (required for both tasks E and F), such as combinatorial semantics, syntax or working memory, may have driven the dissociation of Factor 2 in at least some participants. Another possibility, which we cannot rule out, is that the difference between factors was not linguistic in nature, but driven by specific cognitive requirements of the tasks, e.g. the extent to which they involve executive functions, or simply the overall cognitive demand of the task.” (p. 22-23)

We agree that handedness may be a key factor in helping us understand how language is represented in the brain, and now make this point in a final paragraph:

“One insight from this study is that tasks that do not appear lateralised at the group level may nevertheless show consistent individual differences in strength of laterality. The most striking example was the syntactic comprehension task, which had good test-retest reliability, but was not left-lateralised. Laterality indices from this task also showed significant correlations with other tasks that were left-lateralised (see Figure 3). This suggests that a focus on language laterality as a binary category may be misleading, as it overlooks stable individual differences in strength of lateralisation that may manifest as left- or right-bias depending on the specific task.” (p. 24)

Minor comments:

1.6: Page 18 last sentence, shouldn’t that read tasks B, C, E and F?

Thank you for spotting this error – it has been corrected as suggested.

1.7: Page 22 middle “whether there whether there”

Thank you, these rogue words have been removed.

Reviewer: 2

This manuscript addresses the question of whether there is more than one dimension (factor) underlying cerebral asymmetry for language, and whether this depends on handedness. The analysis is mathematically sophisticated, and I am not sure I fully understood all of it, but I do have a number of concerns:

2.1: It is not clear how handedness is defined. Some measures are based on performance (e.g., the peg-moving task) and some on preference (e.g., the Edinburgh Handedness Inventory), and these are not perfectly correlated. Both kinds of measure imply a continuum rather than a dichotomy, and there is always a subset who are ambidextrous rather than left- or right-handed. I am not really sure of the wisdom of simply dividing the sample into left- and right-handers, and carrying out parallel analyses—it’s never a clean cut. An alternative approach is simply to use handedness as a variable and include it in a factor analysis, to see how and where it loads.

Treating handedness as a continuous variable (e.g. using EHI) would be an alternative way to approach this analysis. Note, however, that the distribution of EHI scores is highly non-normal, so incorporating it into factor analysis would not be straightforward. We agree with the reviewer that there are many different ways to measure handedness, and that there is often surprisingly poor correlation between different measures (e.g. Buenaventura Castillo, Lynch & Paracchini, 2019). Selecting which measure to use is therefore complicated and could have a big impact on the results. Using self-defined handedness as a binary variable avoids this complication, but of course may be an oversimplification of the handedness dimension.

As we did not originally plan to investigate handedness in this study, we did not measure handedness in our original sample of participants (reported in Woodhead et al., 2019), but only recorded their self-report handedness category. When we decided to extend the study we did recontact our original participants to ask them to complete an EHI questionnaire, but only received a response from around two thirds of the participants. Hence we are unable to include a continuous measure of handedness in the current analysis, though we plan to do so in future work.

2.2: It has long been understood that laterality, including handedness itself, is more variable in left-handers (however defined) and early studies based on the effects of lateralized brain injury typically found no difference in asymmetry between left-handers and the ambidextrous, who tended to be lumped together for comparison with right-handers.

Our participant questionnaire offered three handedness categories: left, right or ambidextrous. No participant chose the ambidextrous option. We have noted this in the Methods as follows:

“Handedness was assessed using a self-report question with three options: left handed, right handed or ambidextrous. No participant selected ‘ambidextrous’.” (p. 6)

2.3: Earlier theories of handedness (Annett, McManus) proposed that left-handedness arose through lack of a genetic disposition to right-handedness, so that left-handedness was the outcome of a random process. This would include some right-handers. In its simplest, single-locus form, this theory probably no longer holds, but the idea that left-handedness (and some right-handedness) arises through chance rather than a genetic effect seems to have persisted in multi-locus models (e.g., McManus et al., 2013—and see the predictions for language lateralization on p. 49 of that article). I’m wondering whether this, rather than some putative two-factor model, might explain the pattern of results (e.g., as shown in Figure 2). Perhaps there is simply an extra wash of randomness over the left-handers, creating more variance and bringing them closer to zero.

The notion that there is more randomness in laterality in left-handers makes sense of many phenomena, and there are aspects of our data that agree with such a view. Our emphasis here, though is on the extent to which LIs from different language tasks load on a common factor, and we do not think the pattern of results can simply be accounted for by greater randomness in left-handers. Perhaps the most persuasive data comes from the correlation matrices in Figure 3: you can see there that the intercorrelations between tasks are at least as high in left-handers as in right-handers.

2.4: A two-factor model is always going to produce a better fit than a one-factor model. Perhaps, as the authors suggest, it is a better fit for left-handers than for right-handers, but I’m not sure what this could mean.

The model comparison takes into account model complexity by reporting the AIC which penalises the two factor model. In left-handers there was a substantial difference in AIC between the 1 and 2 factor models (AIC = 502 and 436 respectively), whereas in right-handers the difference was modest (AIC = 569 and 560).

Our results showed that left-handers had robust evidence in favour of a two-factor model (where language lateralisation has two independent components), whereas right-handers had only equivocal evidence in favour of the two-factor model. This indicates a possible difference between the groups in terms of the number of independent components that underlie language lateralisation, but unfortunately a direct comparison between the groups is not possible in this form of analysis (see comment 1.4 above).

Our interpretation is that dissociation between laterality for different factors is more common in left- than right-handers.

2.5: I found it hard to read meaning into the patterns of factor loadings. Did the authors consider rotating axes for a clearer picture of what the different factors might mean? This is fairly standard in factor analytic studies. Or perhaps try to rotate the two solutions into congruence with one another.

Rotating axes is typically performed with exploratory factor analysis. In this study, we use

confirmatory factor analysis and only compare two specific models. We have clarified that we used CFA rather than EFA in the Methods:

*“We used Structural Equation Modelling (SEM) to perform confirmatory factor analysis...”
(p.8)*

*“As our previous paper had obtained good fit for a two-factor model, here we used confirmatory factor analysis, using the same two-factor model, to ask whether left and right handers have different patterns of covariance in language laterality across the six tasks”
(p.8-9)*

We also speculate about the meaning of the patterns of loadings in the Discussion.

2.6: I'm not entirely clear how the factor analysis were done, but factor scores are only crudely estimated by multiplying original scores by factor loadings. In common factor space, where it is assumed that each variable (here, laterality index) has unique variance not present in the factor space, this problem has no exact solution. For an old article on this, see Horn (1965).

We have clarified how we have derived the weighted scores from factor loadings, and changed the terminology to avoid confusion. The approach we adopted corresponds to what DiStefano, Zhu, and Mîndrilă (2009) term a 'non-refined' method of extracting factor scores. We have also taken a weighted mean rather than a weighted sum. The advantage of this approach is that the derived scores are on the same scale as the raw laterality indices, and the plot of the data then shows that Factor 1 is more left-lateralised than Factor 2, as well as allowing one to identify which individuals are left or right-lateralised. This is now explained in the text.

“... we computed 'weighted LI means', as a weighted mean score for each factor from all tasks, using factor loadings as weights. This corresponds to what DiStefano, Zhu, and Mîndrilă (2009) termed a 'non-refined' method of extracting factor scores. These weighted LI means, which are on the same scale as the original raw laterality indices, were plotted in a scatterplot.” (p. 11)

Overall, I'm not sure that we gain from this study a coherent picture of the factor structure of cerebral asymmetry for language, or of exactly what the role of handedness might be.

We agree that we haven't answered all the questions about how cerebral asymmetry for language is structured, but we do think our study advances the field in two novel ways. First, it challenges the conventional view that cerebral lateralisation for language is a single unitary dimension. Second, it confirms that handedness affects not just the mean level of lateralisation, but also the variation in laterality across language domains. This is not the last word, but we think it is a good start, especially as findings are proving replicable across samples.

Reviewer: 3

3.1: Handedness. The paper is about handedness. However, nowhere can I see any statement or discussion about how it was measured. Edinburgh? The Bishop peg moving task? Or perhaps

merely self-description? I suspect the latter or we would have been told more. Either way, we need to be told something. That is particularly the case when on p.22 when the paper says, “It may also be worth further investigating how lateralisation varies as function of the strength of manual preference”; so did the present study not have any measure of strength of manual preference? [Note added later: While browsing through the data I spotted that particdat_Bangor has a column called EHI, but there is no equivalent in the A2 data].

See response to 2.1 and 2.2 above. We have now clarified in the Methods section that handedness was categorised according to self-report.

3.2: How were the left-handers recruited? Again very little on finding some relatively unusual individuals. I may be putting 2+2 to make 5, but I did wonder if some of these left-handers were those previously used by Emma Karlsson in her thesis. Again, a clear statement on recruitment would be useful. Actually I would rather like it if they were the same individuals as then there would be both fTCD and fMRI data in the same individuals, which is precious rare.

Data for the Karlsson thesis was collected 1-3 years earlier and many of the students were therefore no longer part of the participation panel or had left campus. Participants who had taken part in previous experiments were not actively recruited. Accordingly, there was very little overlap between participants from the two studies. Left-handers were recruited using posters distributed around campus, and through a student participation panel. We agree that it would be extremely valuable to have fTCD and fMRI data in the same individuals, and is something we are currently working on in a separate project!

We state in the methods that participants were recruited “via local subject panels or poster advertisements’ (p. 5).

3.3: Removal of trials, cases, etc.. At times I did wonder if too many participants and/or observations were being discarded. I can fully accept technical errors, spikes, etc.. Much more concerning for me was that there seemed an obsessive, almost fetishistic, determination to remove outliers, with everything trying to be forced into normal distributions. The normal distribution is convenient, but it is not always the road to analytic salvation. Sometimes outliers are there because they are different and meaningful. That single left-hander in the 2019 study was the bellwether for other possible problems, but it was an outlier. If one is interested in variation, then removing individuals who are too variable is not a good analytic strategy. If it were height that were being analysed then removing outliers means one would never discover either achondroplasia or acromegaly in the population... I did wonder about an analysis/ dataset with as little removed as possible (technical issues aside).

When analysing the fTCD data, we excluded data in the following ways:

- *Individual trials were excluded if:*
 - o *An obvious artifact was observed, e.g. signal drop out*

- *The data contained values below 60% or above 140% of the mean normalised data range: this is standard practice in fTCD analysis, as it is a simple way of identifying trials with extreme values, which are likely to be due to artefacts*
- *Participants failed to respond or gave an inappropriate response: this ensures that the participant was actually performing the task of interest*
- *Only 4.12% of all trials were excluded for these reasons*
- *One task from a participant was excluded if:*
 - *They had less than 10 usable trials for that task: this ensured that there was sufficient data to calculate a reliable average for that task. This applied to a single task for five participants*
 - *The standard error the trial-by-trial LI values for that task was too high: this ensured that the data for that task was reliable, i.e. not highly variable over the course of the experiment. This applied to a single task for five participants*
- *A participant was excluded if they had more than one task excluded for the reasons stated above. This applied to one participant.*

Hence, we do not exclude data in order to make the data fit a normal distribution: we exclude data where there are artefacts; where the participant did not perform the task; or where the data was unacceptably noisy.

By contrast, the 'outliers' we observe in the Structural Equation Modelling results (Figure 6) are not excluded, as we (like the reviewer) are interested in identifying and understanding exceptions to the rule of 'typical' laterality.

3.4: Normality. From the point of view of the replication crisis, the removal of participants because they do not fit some pre-decided analytic model, particularly to do with normal distributions, is perhaps a relative of 'harking', with theoretically relevant decisions being made before the data are analysed (good), but the data being pre-constrained into a mould which is perhaps not appropriate (bad), only explaining some of the data and discarding awkward or difficult cases. In statistical terms, there are good reasons to believe that laterality data are not normally distributed, but probably are mixtures in some way of normal distributions, perhaps symmetrically arranged around zero (see McManus, 1983, *Cortex*, 19: 198-214).

As stated above, we do not remove participants in order to fit an analytic model. We checked normality to test whether the data were appropriately distributed for our statistical model. We did not manipulate it to make it conform.

3.5: Estimating reliability of laterality measures. One of the several joys of this and the 2019 study is to see a serious attention to reliability. At this point it is worth saying that there are three traditional approaches to reliability. Internal consistency (e.g. alpha), test-retest reliability/stability, and stability/equivalence (different versions of a test on different occasions). The latter is not available as yet (presumably fTCD and fMRI on different occasions), but this study provides test-

retest reliability. I'm not sure it is labelled as such but it can be seen in figure 3, separately for R and L handers, as the correlation of A1-A2, B1-B2 etc.. As the paper says, the reliabilities are not good for A. The test-retest correlations are all higher for left-handers than right-handers (e.g. .8 vs .52 for B1-B2) but that presumably reflects the greater variance in the left-handers scores (visible in figure 4). It may well be that the variance would be greater if 'outliers' had not been removed quite so extensively.

We mention the test-retest reliability and its possible association with handedness on p.15. We do not think that the difference in variation between left and right handers is due to the outliers: the participants whose data was removed entirely was left handed; and of the participants who had one task removed, 3 were left handed and 5 were right handed. Hence, the exclusion of these datapoints (due to artefacts, poor task performance or noise) did not favour right or left handers disproportionately.

3.6: Internal consistency. Looking into the code shows that as well as test-retest reliability there are also odd-even measures of internal consistency, only however for the analyses using the 'peak difference' method rather than the 'mean difference' method. I only looked at the peak difference odd-even consistency measures but they are substantially higher than test-retest reliabilities (and I presume the same would be the case for mean difference measures). That is interesting, as I think in previous studies the odd-even measures have been reported as reliability estimates for data similar to these. What is really needed is a more sophisticated structural model which contains odd-even scores nested within test and retest. That should give a better sense of the true reliability of measures such as these, and also allow comparison with previous studies. An interesting question is whether participants change their lateralisation from occasion to occasion, perhaps due to strategy differences.

We agree with the reviewer that the split-half reliability data (using odd and even trials, with the mean LI method) may be of interest to the reader. We have added this to the analysis scripts and the results are now presented in Supplementary Table 1, as shown below. However, we do not think that splitting the data into odd and even trials would benefit the SEM analysis, as each indicator would be calculated from only ~7 trials.

Task	Session	Mean LI (sd)		Split-half Correlation	
		Left Handers	Right Handers	Left Handers	Right Handers
A	1	0.67 (1.26)	1.12 (1.07)	0.64 ***	0.53 ***
	2	0.88 (1.15)	0.91 (1.09)	0.66 ***	0.43 **
B	1	0.35 (1.96)	1.73 (1.52)	0.79 ***	0.60 ***
	2	0.33 (1.73)	1.52 (1.55)	0.75 ***	0.72 ***
C	1	-0.83 (1.63)	0.90 (1.21)	0.88 ***	0.31 *
	2	-0.77 (1.47)	0.80 (1.17)	0.52 **	0.31 *

D	1	1.52 (2.36)	2.82 (1.68)	0.89 ***	0.70 ***
	2	1.37 (2.38)	3.06 (1.68)	0.89 ***	0.64 ***
E	1	-0.87 (2.24)	1.16 (1.78)	0.81 ***	0.75 ***
	2	-0.49 (1.88)	1.27 (1.53)	0.84 ***	0.65 ***
F	1	-0.61 (1.72)	0.04 (1.55)	0.68 ***	0.76 ***
	2	-0.55 (1.68)	-0.08 (1.47)	0.76 ***	0.64 ***

Supplementary Table 1

Group mean (and standard deviation) laterality indices and split-half correlations are reported for the six tasks (A-F) at the two sessions (1-2) for the left and right handed groups. Split-half reliability was calculated using Pearson's correlations between LI values from odd versus even trials (* $p < .05$; ** $p < .01$; *** $p < .001$).

3.7: Mixtures. Mixtures and mixture modelling are mentioned briefly in relation to the Mazoyer et al 2014 study, but never again appear. "It is possible that what we observed as handedness differences in this study reflects the presence of a subset of these 'atypical' participants in the left handed group but not the right handed group." That opens a whole can of worms, not least about the earlier comment on removing all points which are 'outliers'. Of course the phrase, "in the left handed group but not the right handed group" is also problematic, as in the Karlsson Bangor fMRI study it is very clear that there are atypical right-handers, so excluding them here from being atypical does not seem likely. There are also many right-handed patients in the literature with 'crossed aphasia', even if they are proportionately rare. The Carey and Johnstone meta-analysis (your ref 5) makes that clear.

We are also interested in sub-populations of typical and atypical participants within the handedness groups, but the sample size required to adequately fit SEM models that include finite Gaussian mixture models elements are substantial. The modelling paradigm for mixture models of any kind is also best suited to the Bayesian framework which we do not employ here (Richardson and Green, 1997). Although OpenMx appears to have some facility to implement mixture modelling at the factor level, for example a latent profile model; it does not appear to have the facility to implement mixture at both the measurement variable (manifest) and factor level which may be necessary to properly capture the reviewer's thoughts.

We agree that 'atypical' laterality can also be observed in right handers, although Mazoyer et al (2014) did not find any such participants. To acknowledge this, we have amended the wording on p. 23 as follows:

"It is possible that what we observe as handedness differences in this study reflect the presence of a subset of 'atypical' participants, who are usually (but not always) left handed (Mazoyer et al., 2014; Carey & Johnstone, 2014; Karlsson et al., 2019)."

3.8: The nature of the measurement scale in laterality studies. This is perhaps my major theoretical criticism of the present study, and it eventually comes back to a) mixture distributions and b) the different approaches of the Karlsson and Oxford studies. To start with though we need to get to grips with what is being measured and what laterality is all about.

- a. Handedness, as is clear to anyone, particularly in EHI scores, is not normally distributed. Other lateralities are also clearly not normally distributed – consider the side of the heart in the chest, which is mostly varies in distance to the left of the midline, but in situs inversus varies in distance to the right of the midline. The distribution is bimodal, with the modes placed symmetrically around zero. Something similar applies to the EHI, although the mode of the left-handers is closer to zero than the mode of the right-handers, and the data are clearly censored at the extremes. Skills such as the Tapley-Bryden task are much more obviously bimodal normal (Laterality, 2016, 21: 371-396) with symmetric modes.
- b. The present study though considers a continuum of scores, explicitly assumed to be normally distributed. The dependent variable is the average left-sided blood flow minus the average right-sided blood flow, with zero indicating no difference. Most but not all of the differences are positive, with some being negative. Although that is what the data actually are, there are few places in the analyses where zero plays any special role. They are there in figure 2, but I had to draw them in for figure 4. The SEM analyses though simply know nothing about zero and the fact that >0 means “on the left” and <0 means “on the right”. It is worth remembering that Broca didn’t say that language on average was on the left, but in individuals he said it was either in the left or in the right (just as the heart is not on average on the left side in any meaningful sense). To emphasise the point, if the data were, say, reaction times, then the models would all look identical to their present form. So where does lateralisation really come into these models?
- c. The answer to the above question is, I think, that it doesn’t. But that is to miss out on the key aspect of lateralisation – that it is lateralised...! Here I think is the key way in which the Karlsson and Oxford approaches are so different. Oxford treats lateralisation as a normally distributed continuum, whereas Karlsson treats it ultimately as binary (and for fMRI data that is seen in her very important table 3.11 where for five tasks, 24 out of the 32 possible combinations are found; and for the 64 possible combinations with handedness included, 30 combinations were found, and that was based on just 67 participants). Comparing right and left-handers, of 24 right-handers, 13 (54%) had no atypical lateralities and 1 (4%) had all five as atypical, whereas in 43 left-handers, only 4 (9%) had all five lateralities as typical, and 3 (7%) had all five lateralities as atypical. The analyses of the current study can’t really get at such phenomena at all, although it is possible to ask whether mean scores are significantly different from zero, but little is done with those differences.
- d. Immediately the question is raised as to whether the Karlsson and Oxford approaches are incompatible, or instead are talking about similar things from entirely separate perspectives. If laterality can be considered as a bimodal mixture distribution (back to that Cortex paper) then there are separate questions concerning what we can call direction of lateralisation and degree of lateralisation. Karlsson is concerned with direction and Oxford with degree. Both are valid, but they answer different questions in different ways. Simply analysing overall

scores also confounds direction and degree and can be very difficult to interpret (see the toy example in figure 1 of McManus 1983). The SEM modelling here is unable to separate direction and degree.

- e. It is possible to look at the present data in a manner similar to that of Karlsson, simply by dichotomising the LI score for each task at zero. Very interestingly the code for the statistical analysis, at lines 1273 to 1284, asks “Does the number of left lateralised tasks per participant vary with handedness?”. The answer, just as with Karlsson’s data, is a resounding, Yes. Overall, of 74 participants, only 24 had all six language tasks left lateralised; and indeed 4 had none of them left-lateralised, the other 46 having 1 to 5 not left-lateralised. Some of that may be measurement error, but it is unlikely all to be so. In relation to handedness, $20/43 = 46.5\%$ right-handers and $4/31 = 12.9\%$ left-handers had all six tasks left-lateralised. The Karlsson and Oxford studies are very different in many ways, but the pattern of differences in direction of lateralisation is broadly very similar. I’ve played around with matrices of tetrachoric correlations from the two studies and there is a lot of similarity in the Karlsson fMRI data and the Oxford fTCD data.
- f. What is it that SEM can’t do that analysing direction of laterality can? Essentially the difference is that SEM treats data as being on a continuum, but it has no sense that zero has any substantive meaning at all, beyond being a value between +1 and -1. It is as if one were measuring temperature and ignored the fact that 0 has an important meaning on the Centigrade scale. For analysing air temperature or the length of railway tracks that might not matter very much, but for water there is a key change at zero (which is why zero is set there), and there is a qualitative difference between +1 (liquid water) and -1 (solid ice). There is what physicists call a phase change at 0, and everything is different on one side of 0 and the other. Probably cerebral lateralisation can be construed in the same way, with a phase shift at L-R differences of zero. Any analysis of lateralisation needs to recognise that.
- g. The answer is probably to fit mixture distributions within SEM, which could handle the sort of mixture distributions and phase changes suggested above, but I wouldn’t recommend it for the present study.

What are the implications of the previous section for this manuscript? I don’t think that the paper needs more analyses which take direction and degree into account (and that will provide plenty of material for future work). However the paper does need to discuss differences in direction and degree. That is really necessary because of the seemingly very different approaches of the Karlsson and Oxford groups, and the present paper, not least because of Emma being an author, needs some form of consideration of where the studies are similar and where they are different. Also needed is to recognise the limitations of the current SEM modelling for handling issues of direction and degree. Taking that further can await a later paper.

On the one hand, we remain intrigued by the question of whether the true underlying model should be binary rather than continuous, and we do plan in future studies to do more modeling to test different types of model. And indeed, as our data are open, others who wish to do so are welcome to try other approaches (though we think this dataset is not suitable for testing mixture models as it is too small). However, this paper – already heavy going for those who are not familiar with SEM,

would become indigestible with further analysis. And we would strongly defend SEM, which has provided quite novel insights.

SEM has not, to our knowledge, previously been used to analyse cerebral lateralisation. Contrary to the reviewer, we think that our data indicate that an predominant focus on direction of lateralisation may have held the field back, whereas the conceptualisation of laterality as a continuum has provided novel insights. The most surprising feature of our data is the existence of strong correlations, for continuous data, between LIs that are quite different in the strength of laterality. Consider, for instance Figure 2, which shows that in right-handers, task E is significantly left-lateralised and task F is not lateralised. Nevertheless, the two LIs are strongly correlated (see heatmap of correlations). A similar story obtained with left-handers where task B is not lateralised at the group level, but is correlated around .6-.7 with task D, which is the most lateralised task. This suggests that there are underlying individual differences in strength of lateralisation, but that for some tasks there is a shift to one side. Individuals largely retain their rank ordering on the continuum, but the proportion who are left- lateralised varies from task to task. If we focused on direction of lateralisation, an individual would appear to have inconsistent laterality between the pairs of tasks noted above, whereas with continuous data we see the underlying consistency.

3.9: Left hemisphere functions? The entire paper is about 'language lateralisation' (title) and the implicit assumption is that all of the tasks are left lateralised. That is a bit hard to reconcile however with the overall means, which for SemDec are small and not significantly different from zero, and particularly for Jabber, where the overall mean is clearly negative (-.24), with se of .18, and 40/74 (54%) participants right-lateralised. Are these bilateral tasks, in some sense, or do they have right-hemisphere components to them? Needless to say it would be interesting to know what they looked like with fMRI. I am not sure there is any comment in the paper on these particular measures, but it is perhaps needed. They do correlate with the other measures, but I am not sure that clarifies things. If Knecht et al had started with a Jabberwocky task they might well have abandoned fTCD... At that point I did wish that Knecht et al's word generation task had been included, particularly as it has been called, "the gold standard task for assessing language lateralisation" (Laterality, 2012, 17: 694-710).

In the 2019 paper, we set out to investigate whether all language functions were similarly lateralised, and covaried together, without making the assumption that they would all be left lateralised. The fact that some language tasks are not lateralised at the population level is one of the points that we are making, and so we agree with the reviewer. The reviewer states: "They do correlate with the other measures, but I am not sure that clarifies things", but, as we argue above, this is a key observation that clarifies that one needs to distinguish between differences in group mean laterality for a task, and individual differences in laterality strength, because the latter can be associated for tasks that differ in the former. We have now added to the end of the discussion the following paragraph:

"One insight from this study is that tasks that do not appear lateralised at the group level may nevertheless show consistent individual differences in strength of laterality. The most striking example was the syntactic comprehension task, which had good test-retest reliability, but was not left-lateralised. Laterality indices from this task also showed

significant correlations with other tasks that were left-lateralised (see Figure 3). This suggests that a focus on language laterality as a binary category may be misleading, as it overlooks stable individual differences in strength of lateralisation that may manifest as left- or right-bias depending on the specific task.” (p. 24)

Regarding the word generation task; our selection of tasks was made on theoretical grounds, and on the basis of prior research. We used sentence generation instead because our previous work had shown that it had stronger LI values than word generation, although the two correlated closely (Woodhead et al., 2018). Of course, there are pros and cons for all tasks, and it’s never possible to test everything in one study. If it is of any consolation to the reviewer, we are planning to use word generation in our next project.

We also agree that it would be interesting to see the pattern of activation (or lateralisation) across the whole brain for these tasks using fMRI. With regards to syntactic decision, we used similar stimuli to Goucha & Friederici (2015, Experiment 1), who used fMRI but with a slightly different task design. Whereas we used a two-alternative forced choice sentence/non-sentence decision, they presented the sentences and non-sentences in different blocks, and contrasted the two to identify areas that were more activated by sentences than non-sentences. They used an incidental target word detection task to ensure attention to the stimuli – the participants were not explicitly instructed to focus on syntax. Their sentence>non-sentence contrast revealed activity in the left IFG (BA44 and 45) and left posterior STS.

3.11: Finally, the title. Is it just me, or does the title utterly underplay the importance and interest of this study? It sounds like a mere scholarly footnote, perhaps adding a few more cases, instead of addressing the key issue in the 2019 study which was the lack of left-handers in reasonable numbers. Maybe something like, “The multidimensional structure of language lateralisation: An fTCD study of six language tasks on two occasions in 31 left-handed and 43 right-handed adults”. You get the idea.

The title was chosen as it reflected the fact that this is an Update paper, rather than reporting an entirely new dataset. However, a more informative title may be helpful for orientating the reader. We have enquired with the editor about whether it is possible to reword the title. If so, we’d suggest the alternative title:

“An updated investigation of the multidimensional structure of language lateralisation in left and right handed adults: a test-retest functional transcranial Doppler sonography study with six language tasks”

Minor comments

3.12: p.5. “In order to mitigate the potential confound of testing site, we also aimed to test at least 10 right handed participants at the new site (Bangor)”. Are there comparisons of the Oxford and Bangor data? If so, I couldn’t find them. And is 10 right-handers in Bangor really sufficient?

We don't directly currently compare Oxford and Bangor data, but we have added a pirate plot to the Supplementary Materials, indicating which data was acquired at Oxford or Bangor. We think that this demonstrates that there was no systematic difference between data acquired at the two sites, without the need for formal analysis.

Supplementary Figure 2

Pirate plot showing the laterality indices (LI values) for left and right handed groups, across all tasks (A-F) and both sessions (1, top; and 2, bottom). Black dots represent LI values for each participant tested at Oxford, and red dots denote participants tested at Bangor. A = List Generation; B = Phonological Decision; C = Semantic Decision; D = Sentence Generation; E = Sentence Comprehension; F = Syntactic Decision.

3.13: Handling of missing data in the SEM. Looking carefully at the code, I think it is the case that missing values were handled by means of FIML, full-information maximum likelihood. That is good, but I am not sure it is ever explained in the text. For those less experienced in SEM, there may well be confusion as figure 3 shows correlation matrices, but these presumably have different Ns in the various cells. Much basic SEM, pre-FIML, uses covariance or correlation matrices, and it was tempting at first to assume that you had analysed the matrices presented in figure 3. Analysing correlation matrices would have been problematic since the measures are on the same measurement scale (mean L-R difference in blood flow) and therefore it would have been appropriate here to use covariance matrices rather than correlation matrices, which have all sorts of inherent problems. I can see the expository value of the correlation matrices, but clarification is needed of what you actually did. There may even be an argument for presenting covariance matrices and just accepting it.

To specifically address the question regarding whether we used correlation matrices for input into the SEM, we did not and used the raw data. The correlation matrices were purely for descriptive

purposes to give the reader a clearer picture of the data. We agree with the reviewer that using correlation matrices for SEM input with different Ns can be problematic.

With regards to missing data, we thank the reviewer for pointing out this detail and have added the following sentences to the SEM Methods (p. 10) to aid clarity:

“Missing data was dealt with using the Full-information maximum likelihood (FIML) estimation and the raw data was inputted into the model rather than correlation or covariance matrices. (Note: Correlation matrices for the two groups are presented in Figure 3).”

3.14: You are using SEM models where means are estimated. However the diagram in figure 1 does not reflect that. The diagrams in figure 4 of the 2019 paper make it clearer that both means and variances are being analysed, although I confess that I find it less than clear when means are presented in one diagram and variances in another. Typically both are presented in the same diagram (see the introductory material on OpenMx in <https://openmx.ssri.psu.edu/sites/default/files/OpenMx%20beta%20User%20Guide.pdf> where, say, section 2.2.2 on p.65 presents a clear model with both means and variances). For the present case where there are more variables the solution is probably to have just three variables [A1, B1,...F1) and keep the model easier to read.

We kindly choose to disagree that this is “typical” practice. Means are usually omitted in most scientific publications that present path diagrams. The means are omitted precisely for the reason the reviewers suggests, as it can make the diagram less clear. The purpose of including the means in the 2019 study diagrams was that we started by testing specific hypotheses about means. However, once the focus turned to testing models of covariances, we dropped means from the model diagrams. These latter models are the ones that carry forward to the present study, hence the path diagrams are in correspondence. We have added a note to the Figure legend to make interpretation of the figure easier, and to explain that means were included in the model but omitted from the diagram for simplicity.

“Figure 1

Diagrams for the SEM models tested in this analysis. The letters A-F denote the task, and the numbers 1-2 denote the session. By convention, factors are shown in ovals, and factor loadings are shown with arrows labelled with lower case letters. The circular arrows represent residual variance around the factors or task means. The task means were included in the models, but have been omitted from the model diagrams for simplicity as our analyses focused on comparing the covariance structure (i.e. the one factor model versus the two factor model).” (p. 9)

3.15: Model Fit Statistics. Model Fit Statistics in the 2020 and 2019 papers are presented very differently. Degrees of freedom are complicated, and are likely to confuse the beginner (and it is suggested that most of those interested in laterality are beginners as it is so rare). DF are easiest to

understand when modelling covariance matrices. A 12x12 covariance matrix similar to that in figure 3, has 12 diagonal elements and 66 off-diagonal elements, making 78 df. If means are included there are another 12 elements, making 90 df in total. Models then consume various of those dfs, leaving the others in a residual, which in simple modelling is the df of the chisquare goodness of fit statistic. At that point, it becomes confusing in the 2019 paper when the table says there are 409 df for the fully saturated model (because of FIML). Various of those df do get eaten up by parameters. Constraining some of the parameters, as with the task effect model then gives another 12 df and so on. None of that is obvious. Less obvious still is that table 4 in the 2020 paper doesn't mention df, but instead describes parameters, and gives delta df. That is clearer. The table though has AIC and not BIC as in 2019, the CFIs look nothing like the 2020 values, and there are chi-square statistics with no dfs. The 2020 table has no chisquare statistics or associated p values. This doesn't feel like an 'update' and it also feels rather confusing even when one has used SEM for many years. Some consistency and explanation would probably help those not used to SEM.

There is not necessarily a correspondence between fit of the 2019 and 2020 data (one being a subset of the other). When comparing AIC or BIC, (both likelihood derived indexes) we would not necessarily expect the values to be the same or close, as the likelihood is different in terms of its variance. The same argument applies to the relative model fit indices specifically the CFI, as the same model but with more data may change the goodness of fit as a result of the new data. This is due to the CFI being based on the Chi square statistic that is itself highly sensitive to sample size (Wang & Wang, 2012). Making them directly comparable is difficult as the difference can be attributed to the new data added to the existing data.

Models reported in the 2020 study are only a subset of the models reported in the 2019 paper (Person effect model and Task x person effect model). The final model reported is a multigroup model to directly compare the left and right handers now possible with the increase sample size in left handers.

We have updated Table 4, by adding the BIC values for the 2020 models to permit comparison and reporting the df and likelihood ratio test values rather than differences to ease interpretation.

“Table 4

Model fit statistics for the one and two factor models estimated the left handed and right handed groups separately; and the constrained and unconstrained two factor multigroup models. N params, number of parameters; -2logL , -2 log likelihood; df, degrees of freedom; AIC, Akaike's Information Criterion; BIC, Bayes Information Criterion; CFI, Comparative Fit Index; RMSEA, root mean square error of approximation.

Group	Model	N params	-2logL	df	P-value	AIC	BIC	CFI	RMSEA
Left	1 Factor	18	1204.239	351	-	502.24	-1.09	.727	.204
	2 Factor	23	1128.909	346	7.93e ⁻¹⁵	436.91	-59.25	.933	.105
Right	1 Factor	18	1555.086	493	-	569.09	-299.19	.886	.102

	2 Factor	23	1536.077	488	0.0019	560.08	-299.39	.936	.079
Multi-group	Constrained	36	2688.054	844	-	1000.05	-940.47	.913	.071
	Unconstrained	46	2664.986	834	0.0105	996.99	-924.60	.934	.064

* These fit indices cannot be extracted for the multigroup model via UMX (umx website 2019-07-28). See <https://tbates.github.io/advanced/1995/03/15/detailed-Multigroup.html> for details." (p. 18)

References

- Buenaventura Castillo, C. E., Lynch, A. G., & Paracchini, S. (2019, September 26). Different laterality indexes are poorly correlated with one another but consistently show the tendency of males and females to be more left- and right- lateralised, respectively. <https://doi.org/10.31234/osf.io/he6j2>
- DiStefano, C., Zhu, M., & Mîndrilă, D. (2009). Understanding and using factor scores: Considerations for the applied researcher. *Practical Assessment, Research, and Evaluation, 14*.
doi:<https://doi.org/10.7275/da8t-4g52>
- Horn, J.L. (1965). An empirical comparison of methods for estimating factor scores. *Journal of Educational & Psychological Measurement, 25*, 313-322.

Appendix D

Response to Reviewers

We would like to thank the Associate Editor and the three reviewers for reviewing the revised version of this paper. Our responses to their comments are shown below.

1. Associate Editor

1.1 One final point which I think requires acknowledgement in the discussion is the issue of the dimensionality of handedness, which is a point raised by two of the reviewers. It's now clear in the ms that you used a dichotomous self-report measure, but as you say in your response to the reviews, a continuous measure of handedness would be an alternative approach, and one which would be interesting to take in future work.

With respect, this point has already been made in our Discussion, but we have added to this paragraph to address Reviewer 2's suggestion:

“It may also be worth further investigating how lateralisation varies as function of the strength of manual preference. Two large fTCD studies have been explored this to date, both using a word generation task: the first (N=326)(3) found a linear relationship, but the second (N=310)(1) found no association. In addition, the relationship between handedness (as a continuous measure), language and gesture processing (praxis) was examined in an fMRI study (N=94) by Häberling and colleagues (40). With regards to language, they observed significant correlations between handedness and LIs for word generation in BA44/45, and for synonym decision in BA44/45 and the temporal lobe. However, using factor analysis across LIs from all praxis and language tasks, they found that handedness did not load significantly on the factor most strongly related to language laterality.” (p.22-23).

2. Reviewer 2

2.1 In spite of the sophisticated statistical analysis I'm just not very convinced. Defining handedness by self report gives rather a crude dichotomy--handedness is really a continuum and I don't know what it really means to apply separate factor analyses to left and right handers. I see no real reason not to rotate axes to seek meaning in the factors, even though it's supposedly confirmatory rather than exploratory. For an alternative factor analytic approach which does treat handedness as a continuous variable, see Häberling, I.S., et al. (2016), Cortex, 82, 72-85. <http://dx.doi.org/10.1016/j.cortex.2016.06.003> 0010-9452, which should at least be cited. To my mind, at least, this provides a more interpretable solution, and I can't easily reconcile it with what is claimed here.

We'd like to thank the reviewer for directing us to this interesting paper, which we have now included in the Discussion as described above. It's a shame that we weren't able to

retrospectively acquire a full handedness inventory from the original group of participants – as mentioned before, we did try to get this, but not all participants responded to our request. It's definitely an interesting approach that we would consider in future studies, as we agree that there may be more to be gained by treating handedness as a continuous measure.

3. Reviewer 3

3.1 My only specific comment, which does not require any further changes, refers to the reply in 3.8 that, "this paper – already heavy going for those who are not familiar with SEM, would become indigestible with further analysis". I can see the point of this, but there is also an argument that if the phenomena require analyses which are heavy-going, then that is where the science has to go, as otherwise the phenomena will not be explored properly. Fortunately most physicists in their primary research papers do not restrict themselves to using only school-level maths or it seems unlikely that gravity-waves or whatever would have been discovered. At some point 'the reader' has to be willing to grapple with the issues, even if they are difficult. End of sermon!

We agree that sometimes complicated concepts require complicated analyses, but we hope that we have found a balance between complexity and readability in this manuscript.